# Aberrant causal inference and presence of a compensatory mechanism in autism spectrum disorder

**Jean-Paul Noel[1†], Sabyasachi Shivkumar[2†], Kalpana Dokka[3†], Ralf M Haefner[2‡], Dora E Angelaki[1,3*‡]**

[1]Center for Neural Science, New York University, New York City, United States; [2]Brain and Cognitive Sciences, University of Rochester, Rochester, United States; [3]Department of Neuroscience, Baylor College of Medicine, Houston, United States

**Abstract** Autism spectrum disorder (ASD) is characterized by a panoply of social, communicative, and sensory anomalies. As such, a central goal of computational psychiatry is to ascribe the heterogenous phenotypes observed in ASD to a limited set of canonical computations that may have gone awry in the disorder. Here, we posit causal inference – the process of inferring a causal structure linking sensory signals to hidden world causes – as one such computation. We show that audio-visual integration is intact in ASD and in line with optimal models of cue combination, yet multisensory behavior is anomalous in ASD because this group operates under an internal model favoring integration (vs. segregation). Paradoxically, during explicit reports of common cause across spatial or temporal disparities, individuals with ASD were less and not more likely to report common cause, particularly at small cue disparities. Formal model fitting revealed differences in both the prior probability for common cause (p-common) and choice biases, which are dissociable in implicit but not explicit causal inference tasks. Together, this pattern of results suggests (i) different internal models in attributing world causes to sensory signals in ASD relative to neurotypical individuals given identical sensory cues, and (ii) the presence of an explicit compensatory mechanism in ASD, with these individuals putatively having learned to compensate for their bias to integrate in explicit reports.

**\*For correspondence:**
da93@nyu.edu

[†]These authors contributed equally to this work
[‡]These authors also contributed equally to this work

**Competing interest:** The authors declare that no competing interests exist.

## Editor's evaluation

Autism spectrum disorder is characterized by social, communicative and sensory anomalies. This study uses behavioral psychophysics experiments and computational modelling to interrogate how individuals with autism combine sensory cues in multisensory tasks. The results showed that individuals with autism were more likely to integrate cues, but less likely to report doing so, thus raising interesting questions regarding how individuals with autism perceive the world.

## Introduction

Autism spectrum disorder (ASD) is a heterogenous neurodevelopmental condition characterized by impairments across social, communicative, and sensory domains (*American Psychiatric Association, 2013*; see also *Robertson and Baron-Cohen, 2017* for a review focused on sensory processing in ASD). Given this vast heterogeneity, many *Lawson et al., 2017*; *Lawson et al., 2017*; *Lawson et al., 2014*; *Lieder et al., 2019*; *Noel et al., 2020*; *Noel et al., 2021a*, *Noel et al., 2021b*; *Series, 2020* have recently turned their attention to computational psychiatry to ascribe the diverse phenotypes within the disorder to a set of canonical computations that may have gone awry.

A strong yet unexplored candidate for such a computation is causal inference (*Körding et al., 2007*). In causal inference, observers first make use of observations from their sensory milieu to deduce a putative causal structure – a set of relations between hidden (i.e. not directly observable) source(s) in the world and sensory signals (e.g. photons hitting your retina and air-compression waves impacting your cochlea). For instance, in the presence of auditory and visual speech signals, one may hypothesize a single speaker emitting both auditory and visual signals, or contrarily, the presence of two sources, e.g., a puppet mouthing (visual) and the unskillful ventriloquist emitting sounds (auditory). This internal model linking world sources to signals then impacts downstream processes. If signals are hypothesized to come from a common source, observers may combine these redundant signals to ameliorate the precision (*Ernst and Banks, 2002*) and accuracy (*Odegaard et al., 2015*; *Dokka et al., 2015*) of their estimates. In fact, an array of studies *Ernst and Banks, 2002*; *Hillis et al., 2002*; *Alais and Burr, 2004*; *Kersten et al., 2004* have suggested that humans combine sensory signals weighted by their reliability. On the other hand, hypothesizing that a single source exists, when in fact multiple do, may lead to perceptual biases (as in the ventriloquist example).

It is well established that humans perform causal inference in solving a wide array of tasks, such as spatial localization (*Körding et al., 2007*; *Odegaard et al., 2015*; *Rohe and Noppeney, 2015*; *Rohe and Noppeney, 2016*), orientation judgments (*van den Berg et al., 2012*), oddity detection (*Hospedales and Vijayakumar, 2009*), rate detection (*Cao et al., 2019*), verticality estimation (*de Winkel et al., 2018*), spatial constancy (*Perdreau et al., 2019*), speech perception (*Magnotti et al., 2013*), time-interval perception (*Sawai et al., 2012*), and heading estimation (*Acerbi et al., 2018*; *Dokka et al., 2019*), among others. As such, causal inference may be a canonical computation, ubiquitously guiding adaptive behavior and putatively underlying a wide array of (anomalous) phenotypes, as is observed in autism.

Indeed, the hypothesis that causal inference may be anomalous in ASD is supported by a multitude of tangential evidence, particularly within the study of multisensory perception. Namely, the claims that multisensory perception is anomalous in ASD are abundant and well established (see *Baum et al., 2015* and *Wallace et al., 2020*, for recent reviews), yet these studies tend to lack a strong computational backbone and have not explored whether these deficits truly lie in the ability to perform cue combination, or in the ability to deduce when cues ought to (vs. not) be combined. In this vein, we have demonstrated that optimal cue combination for visual and vestibular signals is intact in ASD (*Zaidel et al., 2015*). In turn, the root of the multisensory deficits in ASD may not be in the integration process itself (see *Noel et al., 2020*, for recent evidence suggesting intact integration over a protracted timescale in ASD), but in establishing an internal model suggesting when signals ought to be integrated vs. segregated – a process of causal inference.

Here we employ multiple audio-visual behavioral tasks to test the hypothesis that causal inference may be aberrant in ASD. These tasks separate cue integration from causal inference, consider both explicit and implicit causal inference tasks, and explore both the spatial and temporal domains. Importantly, we bridge across these experiments by estimating features of causal inference in ASD and control individuals via computational modeling. Finally, we entertain a set of alternative models beyond that of causal inference that could in principle account for differences in behavior between the ASD and control cohorts and highlight which parameters governing causal inference are formally dissociable in implicit vs. explicit tasks (these latter ones constituting a large share of the studies of perceptual abilities in ASD).

## Results

### Intact audio-visual optimal cue integration

First, we probe whether individuals with ASD show a normal or impaired ability to optimally combine sensory cues across audio-visual pairings. To do so, individuals with ASD (n=31; mean ± S.E.M; 15.2±0.4 years; 5 females) and age-matched neurotypical controls (n=34, 16.1±0.4 years; 9 females) viewed a visual disk and/or heard an audio beep for 50 ms. The auditory tone and visual flash were synchronously presented either at the same location (*Figure 1A*, left panel) or separated by a small spatial disparity Δ = ±6° (*Figure 1A*, right panel). The disparity was small enough to escape perceptual awareness (see explicit reports below for corroboration). The auditory stimulus was always the same, making the auditory signals equally reliable across trials. The reliability of the visual cue was

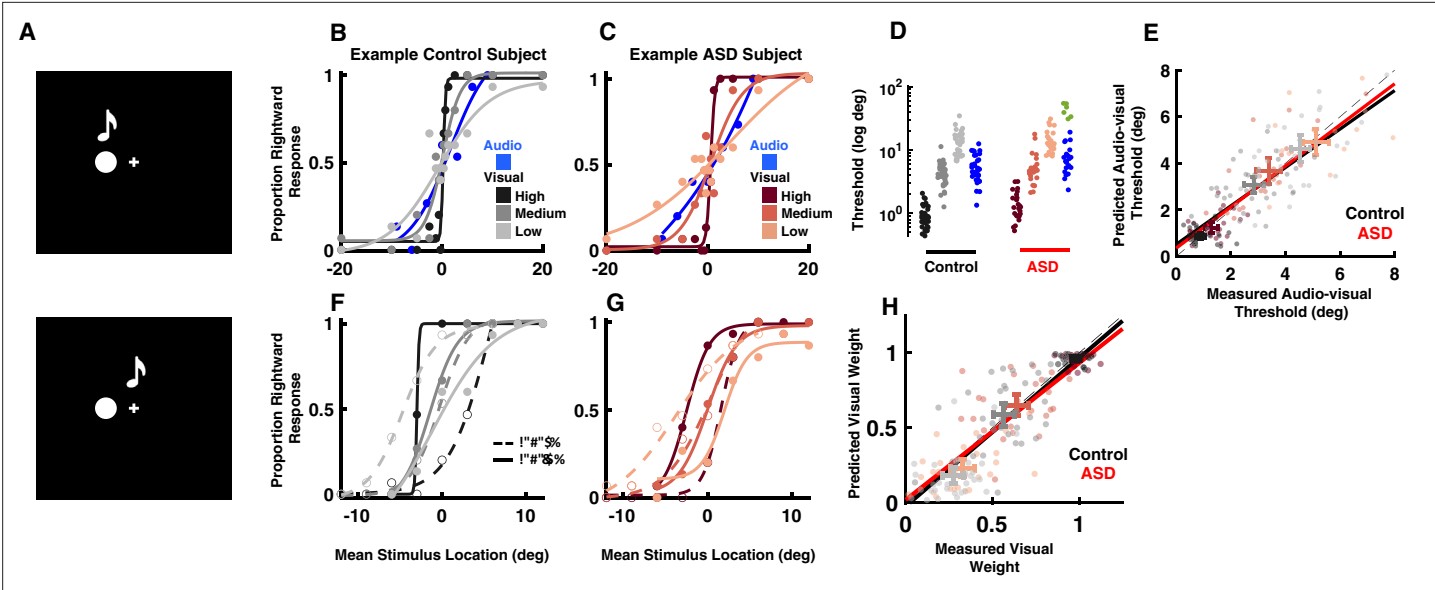

**Figure 1.** Audio-visual optimal cue combination in autism spectrum disorder (ASD). (**A**) Participants (neurotypical control or individual with ASD) viewed a visual disk and heard an auditory tone at different locations and with different small disparities (top = no disparity, bottom = small disparity). They had to indicate the location of the audio-visual event. (**B**) Rightward (from straight ahead) responses (y-axis) as a function of stimulus location (x-axis, positive = rightward) for an example, control subject. Color gradient (from darkest to lightest) indicates the reliability of the visual cue. (**C**) As (**B**), but for an example, ASD subject. (**D**) Discrimination thresholds in localizing audio (blue) or visual stimuli with different reliabilities (color gradient) for control (black) and ASD (red) subjects. Every point is an individual subject. A subset of six ASD subjects had very poor goodness of fit to a cumulative Gaussian (green) and were excluded from subsequent analyses. (**E**) Measured (x-axis) vs. predicted (y-axis) audio-visual discrimination threshold, as predicted by optimal cue integration. Black and red lines are the fit to all participants and reliabilities, respectively, for the control and ASD subjects. Two-dimensional error bars are the mean and 95% CI for each participant group and reliability condition. (**F**) Rightward response of an example control subject as a function of mean stimulus location (x-axis, auditory at +3 and visual –3 would result in mean stimulus location = 0) and disparity, the visual stimuli being either to the right (solid curve) or left (dashed) of the auditory stimuli. Color gradient shows the same gradient in reliability of the visual cue as in (**B**). (**G**) As (**F**), but for an example, ASD subject. (**H**) Measured (x-axis) vs. predicted (y-axis) visual weights, according to *Equation 2* (Methods). Convention follows that established in (**E**). Both control (black) and ASD (red) subjects dynamically adjust the weight attributed to each sensory modality according to their relative reliability.

manipulated by varying the size of the visual stimulus (see Methods for detail). On each trial, subjects indicated if the stimulus appeared to the right or left from straight ahead.

*Figure 1B and C*, respectively, shows the location discrimination of unisensory stimuli (audio in blue and visual according to a color gradient) for an example, control and ASD subject. Overall, subjects with ASD (6.83±0.68°) localized the visual stimulus as well as neurotypical subjects (6.30±0.49°, *Figure 1D*, no group effect, F[1, 57]=0.88, p=0.35, $\eta$2=0.01). As visual reliability decreased (lighter colors), the psychometric curves became flatter indicating larger spatial discrimination thresholds (high reliability: 1.10±0.07°, medium: 4.76±0.36°, low: 13.96±0.82°). This effect of visual reliability was equal across both subject groups (group × reliability interaction, F[2, 114]=0.11, p=0.89, $\eta$2<0.01), with visual thresholds being equal in control and ASD across all reliability levels. Auditory discrimination seemed to highlight potentially two subgroups within the ASD cohort (blue vs. green). Auditory threshold estimation was not possible for 6 of the 31 subjects within the ASD group (*Figure 1D*, green, $R^2$ value <0.50), due to a lack of modulation in their reports as a function of cue location (excluding these 6 subjects, average $R^2$ neurotypical control = 0.95; average $R^2$ ASD = 0.96). Given that the central interest here is in interrogating audio-visual cue combination, and its agreement or disagreement with optimal models of cue combination, the rest of the analyses focuses on the 25 ASD subjects (and the control cohort) who were able to localize auditory tones. Auditory thresholds were similar across neurotypical controls and the ASD cohort where threshold estimation was possible (t57=–1.14, p=0.21, Cohen's d=0.11).

The central hallmark of multisensory cue combination is the improvement in the precision of estimates (e.g. reduced discrimination thresholds) resulting from the integration of redundant signals. Optimal integration (*Ernst and Banks, 2002*) specifies exactly what ought to be the thresholds

derived from integrating two cues, and thus we can compare measured and predicted audio-visual thresholds, according to optimal integration (see *Equations 1; 2* in *Methods*). *Figure 1E* demonstrates that indeed both control (gradients of black) and ASD (gradients of red) subjects combined cues in line with predictions from statistical optimality (control, slope = 0.93, 95% CI = [0.85–1.04]; ASD, slope = 0.94, 95% CI = [0.88–1.08]). These results generalize previous findings from *Zaidel et al., 2015* and suggest that across sensory pairings (e.g. audio-visual here, visuo-vestibular in *Zaidel et al., 2015*) statistically optimal integration of multisensory cues is intact in ASD.

A second characteristic of statistically optimal integration is the ability to dynamically alter the weight attributed to each sensory modality according to their relative reliability, i.e., decreasing the weight assigned to less reliable cues. *Figure 1F and G*, respectively, shows example psychometric functions for an example control and ASD individual when auditory and visual stimuli were separated by a small spatial disparity (Δ=±6°). Both show the same pattern. When the auditory stimulus was to the right of the visual stimulus (Δ=6°, dashed curves), psychometric curves at high reliability (dark black and red symbols for control and ASD) were shifted to the right indicating a leftward bias, in the direction of the visual cue (see *Methods*). At low visual reliability, psychometric curves shifted to the left indicating a rightward bias, toward the auditory cue. That is, in line with predictions from optimal cue combination, psychometric curves shifted to indicate auditory or visual 'dominance', respectively, when auditory and visual cues were the most reliable. Analogous shifts of the psychometric functions were observed when the auditory stimulus was to the left of the visual stimulus (Δ=−6°, solid curves). At the intermediary visual reliability – matching the reliability of auditory cues (*Figure 1D*) – both stimuli influenced localization performance about equally. Such a shift from visual to auditory dominance as the visual cue reliability worsened was prevalent across ASD and control subjects. Importantly, measured and predicted visual weights according to optimal cue combination were well matched in control (*Figure 1H*, black, slope = 0.97, 95% CI = [0.92–1.02]) and ASD (*Figure 1H*, red, slope = 0.99, 95% CI = [0.93–1.05]) groups. Measured visual weights were also not different between groups at any reliability (F[2, 114]=1.11, p=0.33, $\eta$2=0.02). Thus, just as their neurotypical counterparts, ASD subjects dynamically reweighted auditory and visual cues on a trial-by-trial basis depending on their relative reliabilities. Together, this pattern of results suggests that individuals with ASD did not show impairments in integrating perceptually congruent (and near-congruent) auditory and visual stimuli.

## Impaired audio-visual causal inference

Having established that the process of integration is itself intact in ASD, we next queried implicit causal inference – the more general problem of establishing when cues ought to be integrated vs. segregated. Individuals with ASD (n=21, 17.32±0.57 years; 5 females) and age-matched neurotypical controls (n=15, 16.86±0.55 years; 7 females, see *Supplementary file 1*, *Supplementary file 2* for overlap in cohorts across experiments) discriminated the location of an auditory tone (50 ms), while a visual disk was presented synchronously at varying spatial disparities. The stimuli were identical to those above but spanned a larger disparity range (Δ=±3,±6,±12, and ±24°), including those large enough to be perceived as separate events (see explicit reports below). Subjects indicated if the auditory stimulus was located to the left or right of straight ahead, and as above, we fit psychometric curves to estimate perceptual biases. The addition of large audio-visual disparities fundamentally changes the nature of the experiment, where now observers must first ascertain an internal model, i.e., whether auditory and visual cues come from the same or separate world sources. As the disparity between cues increases, we first expect to see the emergence of perceptual biases – one cue influencing the localization of the other. However, as cue disparities continue to increase, we expect observers to switch worldviews, from a regime where cues are hypothesized to come from the same source, to one where cues are now hypothesized to come from separate sources. Thus, as cue disparities continue to increase, eventually the conflict between cues ought to be large enough that perceptual biases asymptote or decrease, given that the observer is operating under the correct internal model (*Körding et al., 2007*; *Rohe and Noppeney, 2015*; *Rohe and Noppeney, 2016*; *Rohe et al., 2019*; *Cao et al., 2019*; *Noel and Angelaki, 2022*).

Overall, individuals with ASD showed a larger bias (i.e. absolute value of the mean of the cumulative Gaussian fit) in auditory localization than the control group (see *Figure 2A and B*, respectively, for control and ASD cohorts; F[1, 34]=5.44, p=0.025, $\eta$2=0.13). Further, how the bias varied with spatial disparity (Δ) significantly differed between the groups (group × disparity interaction: F[7, 168]=3.50,

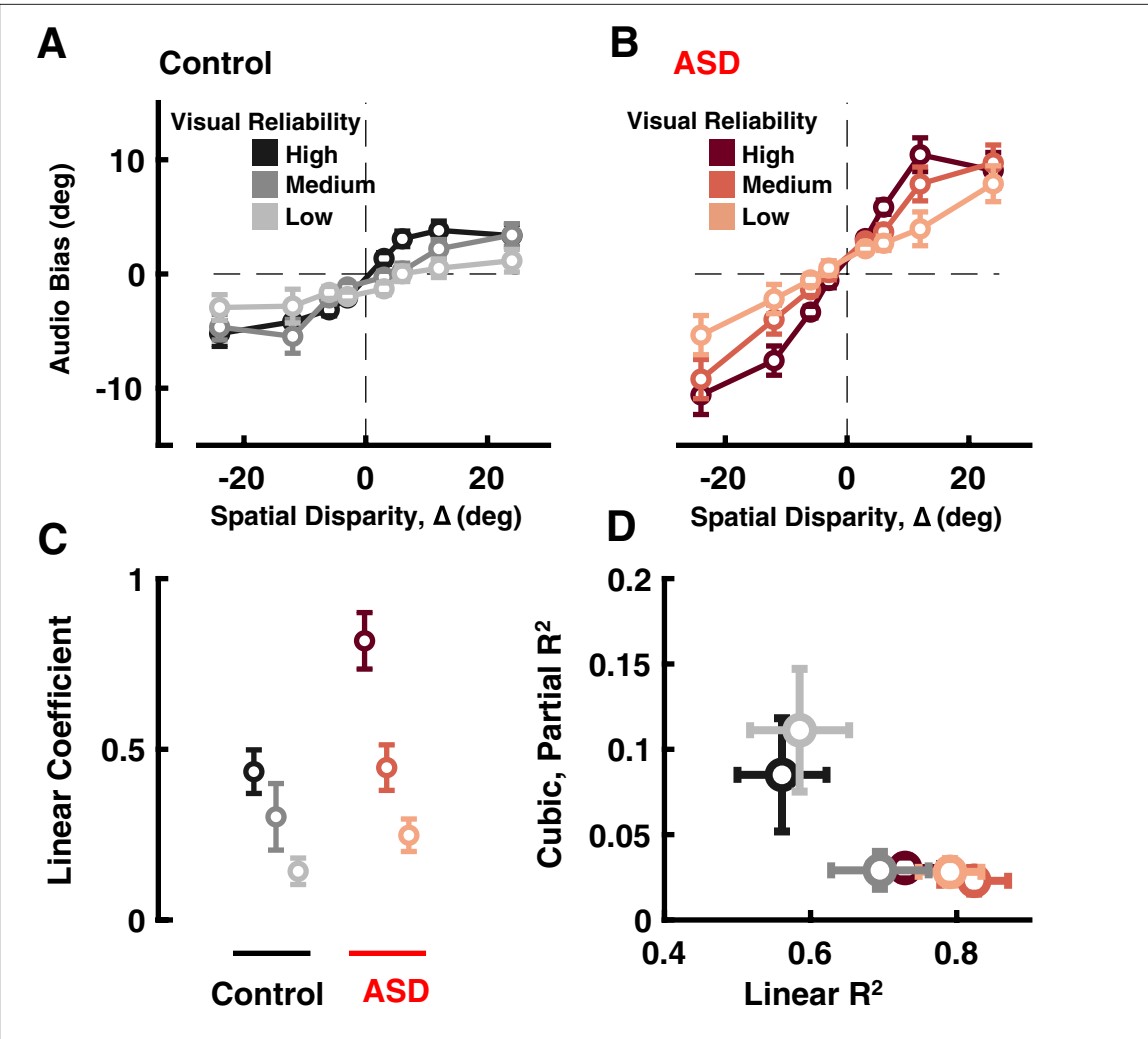

**Figure 2.** Audio-visual causal inference. Participants (black = control; ASD = red) localized auditory tones relative to straight ahead, in the presence of visual cues at different disparities of up to 24°. See *Supplementary file 1*, *Supplementary file 2* for overlap of subjects with *Figure 1*. (**A**) Auditory bias (y-axis, central point of the cumulative Gaussian, e.g. *Figure 1B*) as a function of spatial disparity (x-axis, relative location of the visual cue) and reliability of the visual cue (darker = more reliable) in control subjects. (**B**) As (**A**), but for individuals with ASD. (**C**) Coefficient of the linear fits (y-axis, larger value indicates quicker increase in bias with relative visual location) in control (black) and ASD (red), as a function of visual cue reliability (darker = more reliable). (**D**) Linear $R^2$ (x-axis) demonstrates that the linear fits account well for observed ASD data. On the other hand, adding a cubic term (y-axis, partial $R^2$) improved fit to control data (at two reliabilities) but not ASD data. Error bars are ±1 S.E.M.

The online version of this article includes the following figure supplement(s) for figure 2:

**Figure supplement 1.** Visual and auditory localization performance of participants in Experiment 2 (audio-visual implicit causal inference).

**Figure supplement 2.** Heading discrimination during concurrent implied self-motion and object motion.

---

p=0.002, $\eta 2$=0.12). While the bias saturated at higher Δ in neurotypical subjects, as expected under causal inference, the bias increased monotonically as Δ increased in the ASD group. Thus, despite increasing spatial discrepancy, ASD subjects tended to integrate the cues, as if they nearly always utilized visual signals to localize the auditory cue and did not readily switch to a worldview where the auditory and visual cues did not come from the same world source. The effect of visual cue reliability was similar in both groups (group × reliability interaction, F[2, 168]=1.05, p=0.35, $\eta 2$=0.01), indicating that the auditory bias decreased as visual cue reliability worsened in both groups.

To more rigorously quantify how auditory localization depended on Δ, we fit a third-order regression model to the auditory bias as a function of Δ, independently for each subject and at each visual reliability (y=$a_0$+$a_1$Δ+$a_2$Δ$^2$+$a_3$Δ$^3$; see *Methods*). As shown in *Figure 2C*, across all visual reliabilities,

the ASD group had a larger linear coefficient ($a_1$, ANOVA: F[1, 34]=6.69, p=0.014, $\eta$2=0.16), again indicating a monotonic increase in bias with cue spatial disparity.

To better account for putative non-linear effects at large Δ - those which ought to most clearly index a change from integration to segregation - we fit different regression models (i.e. null, linear, quadratic, and cubic) and estimated the added variance accounted by adding a cubic term (partial $R^2$). This latter term may account for non-linear effects at large Δ, where the impact of visual stimuli on auditory localization may saturate or even decrease ($a_3$ being zero or negative) at large disparities. Results showed that not only the linear term accounted for more variance in the ASD data than controls (*Figure 2D* and x-axis, ANOVA: F[1, 34]=7.08, p=0.012, $\eta$2=0.17), but also the addition of a cubic term significantly improved fits in the control, but not ASD, group (*Figure 2D* and y-axis, partial $R^2$, ANOVA: F[1, 34]=9.87, p=0.003, $\eta$2=0.22). Taken together, these results suggest that contrary to predictions from causal inference – where disparate cues should affect one another at small but not large disparities, i.e., only when they may reasonably index the same source – ASD subjects were not able to down-weight the impact of visual cues on auditory localization at large spatial disparities, resulting in larger errors in auditory localization.

To confirm that the larger biases observed within the ASD cohort were in fact due to these subjects using an incorrect internal model, and not a general impairment in cue localization, we compared unisensory visual and auditory localization thresholds and biases between experimental groups. From the 21 ASD and 15 control subjects who participated in the audio-visual causal inference experiment (Experiment 2), respectively, 15 and 14 of these also participated in Experiment 1 - performing an auditory and visual localization experiment with no disparity (see *Supplementary file 1*, *Supplementary file 2* for further detail). *Figure 2—figure supplement 1A* shows the psychometric functions (auditory localization and visual localization at three different reliability levels) for all subjects participating in Experiment 2. Psychometric thresholds (*Figure 2—figure supplement 1B*, all p>0.09), bias (*Figure 2—figure supplement 1C*, all p>0.11), and goodness of fit (*Figure 2—figure supplement 1D*, all p>0.26) were not significantly different between the ASD and control cohorts, across visual and auditory modalities, and across all reliabilities.

Last, to further bolster the conclusion that individuals with ASD show anomalous implicit causal inference, we replicate the same effect in a very different experimental setup. Namely, subjects (n=17 controls, n=14 ASD, see *Supplementary file 1*, *Supplementary file 2*) performed a visual heading discrimination task requiring the attribution of optic flow signals to self-motion and/or object-motion (a causal inference task requiring the attribution of motion across the retina to multiple sources, self and/or object; see *Dokka et al., 2019*, *Methods*, and *Figure 2—figure supplement 2A* for further detail). We describe the details in the *Supplementary materials* given that the task is not audio-visual and has a different generative model (*Figure 2—figure supplement 2B*). Importantly, however, the results demonstrate that while heading biases are present during intermediate self-velocity disparities and object-velocity disparities for controls and ASD subjects (*Figure 2—figure supplement 2C, D*), they disappear during large cue discrepancies in control subjects, but not ASD subjects. Just as in the audio-visual case (*Figure 2*)**,** ASD subjects do not readily change worldviews and move from integration to segregation as disparities increase (*Figure 2—figure supplement 2C, D*).

Together, these results suggest that in ASD the process of integrating information across modalities is normal (see *Zaidel et al., 2015*) once a correct internal model of the causal structure of the world has been formed. However, the process of inferring this causal structure – the set of relations between hidden sources and sensory signals that may have given rise to the observed data – is anomalous. Namely, individuals with ASD seem to operate under the assumption that sensory cues ought to be integrated most of the time, even for large disparities. Next, we questioned if and how this deficit in causal inference expresses explicitly in overt reports.

## Decreased disparity-independent explicit report of common cause

Individuals with ASD (n=23; 16.14±0.51 years; 5 females) and age-matched neurotypical controls (n=24; 17.10±0.42 years; 7 females; see *Supplementary file 1*, *Supplementary file 2* for overlap in cohorts with previous experiments) viewed a visual disk and heard an auditory tone presented synchronously (50 ms), but at different spatial disparities (same stimuli as above, disparity up to 24°). Participants indicated whether the auditory and visual cues originated from a common source, or from two separate sources (see *Methods* for instructions). In contrast to the localization experiments,

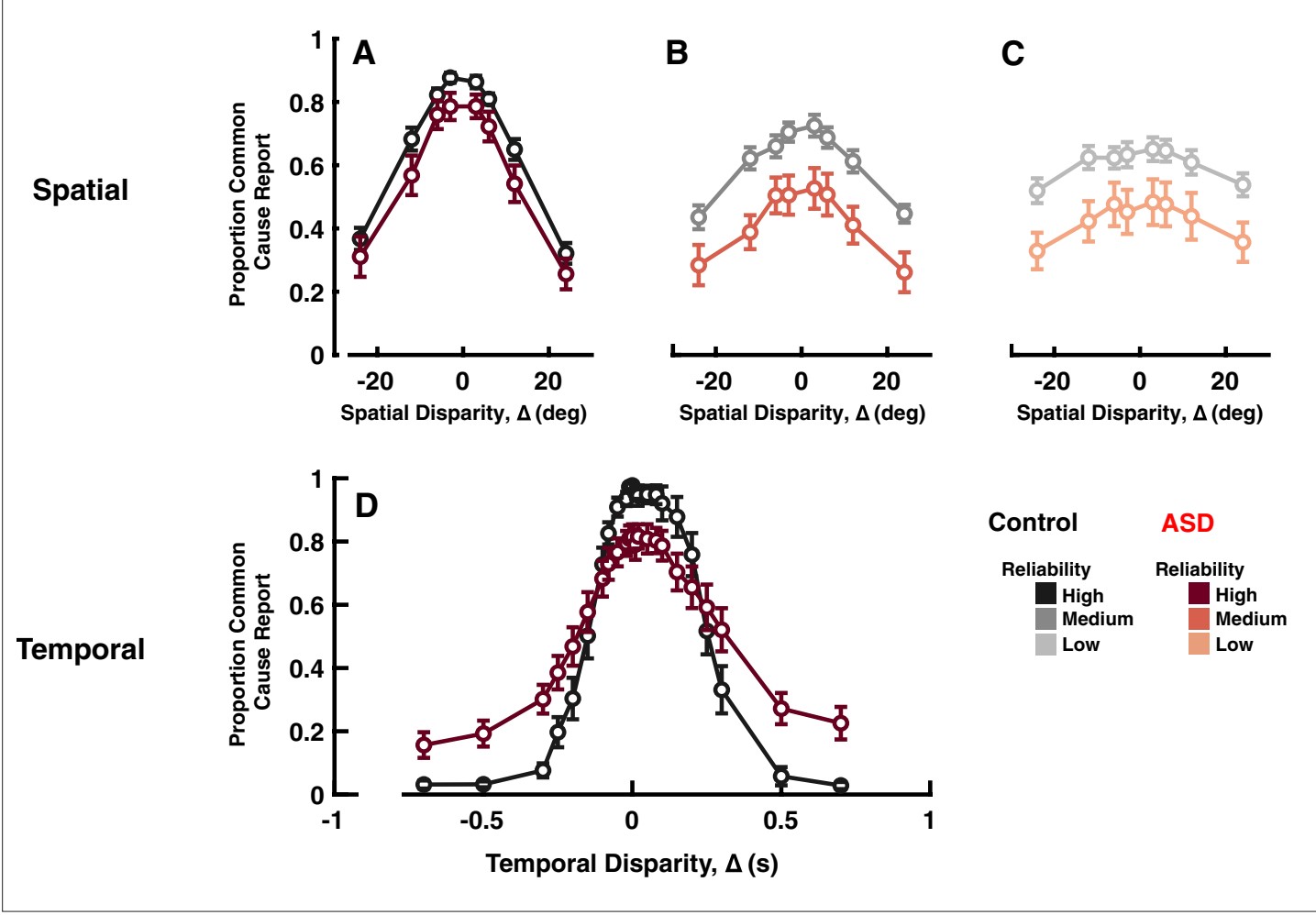

**Figure 3.** Explicit common cause reports across spatial (top) and temporal (bottom) disparities. Proportion of common cause reports (y-axis) as a function of spatial disparity (x-axis) and visual cue reliability; high (**A**), medium (**B**), or low (**C**). The most striking characteristic is the reduced likelihood to report common cause, across any disparity or cue reliability. (**D**) Proportion of common cause reports (y-axis) as a function of temporal disparity. As indexed by many (e.g. *Feldman et al., 2018*) individuals with autism spectrum disorder (ASD) show larger 'temporal binding windows'; temporal extent over which they are likely to report common cause. However, these individuals are also less likely to report common cause, when auditory and visual stimuli are in very close temporal proximity (an effect sometimes reported, e.g., *Noel et al., 2018b*, but many times neglected, given normalization from 0 to 1, to index binding windows; see e.g., *Woynaroski et al., 2013*; *Dunham et al., 2020*). See *Supplementary file 1*, *Supplementary file 2* for overlap of subjects with previous figures. Error bars are ±1 S.E.M.

The online version of this article includes the following figure supplement(s) for figure 3:

**Figure supplement 1.** Visual and auditory localization performance of participants in Experiment 3 (audio-visual explicit causal inference).

**Figure supplement 2.** Reports of common cause as a function of spatial disparity.

**Figure supplement 3.** Fitting a functional form to the explicit causal inference reports.

where subjects localized the physical position of stimuli, here subjects were asked to explicitly report the relationship between the auditory and visual stimuli. See *Figure 3—figure supplement 1* for the unisensory discrimination performance in participants who took part in both the cue integration experiment (Experiment 1) and the current explicit common cause report across spatial disparities. Auditory and visual localization thresholds (all $p > 0.07$), bias (all $p > 0.15$), and the goodness of fit (all $p > 0.16$) of these psychometric estimates were no different between the ASD and control cohort participating in this explicit causal inference judgment experiment.

As expected, most subjects reported a common source more frequently at smaller rather than larger Δ (*Figure 3* $F[8, 259] = 94.86$, $p < 0.001$, $\eta 2 = 0.74$). Interestingly, while this pattern was true for all individual control subjects, eight of the individuals with ASD (i.e. ~⅓ of the cohort) did not modulate

their explicit common cause reports as a function of spatial disparity, despite good auditory and visual localization (see *Figure 3—figure supplement 1* and *Figure 3—figure supplement 2*). These subjects were not included in subsequent analyses. For lower visual reliability (*Figure 3*, from **A-C**), both groups reported common cause less frequently (F[2, 74]=10.68, p<0.001, $\eta$ 2=0.22). A striking difference between experimental groups was the decreased likelihood of reporting common cause, across spatial disparities and visual cue reliabilities, in ASD relative to controls (*Figure 3A–C* shades of black vs. shades of red, F[1, 37]=11.6, p=0.002, $\eta$ 2=0.23). This pattern of results using an explicit causal inference task is opposite from that described for the implicit task of auditory localization, where individuals with ASD were more, and not less, likely to combine cues.

These differences were quantified by fitting Gaussian functions to the proportion of common source reports as a function of Δ (excluding the eight ASD subjects with no modulation in their reports; $R^2$ for this cohort <0.5). The Gaussian fits (control: $R^2$=0.89±0.02; ASD: $R^2$=0.93±0.01) yield three parameters that characterize subjects' behavior: (1) peak amplitude, which represents the maximum proportion of common source reports; (2) mean, which represents the Δ at which subjects perceived a common source most frequently; and (3) width (SD), which represents the range of Δ over which the participant was likely to perceive a common source. Both control and ASD participants perceived a common source most frequently at a Δ close to 0°, and there was no group difference for this parameter (control = 0.30±1.33°; ASD = 0.48±1.9°; F[1, 37]<0.01, p=0.92, $\eta$ 2<0.01). Amplitude and width, however, differed between the two groups. The peak amplitude of the best-fit Gaussian was smaller for the ASD than the control group (control = 0.75±0.02; ASD = 0.62±0.05; F[1, 37]=8.44, p=0.0006, $\eta$ 2=0.18), quantifying the fact that the ASD group perceived a common source less frequently than control participants. The width of the Gaussian fit was smaller in the ASD compared to the control group (control = 30.21±2.10°; ASD = 22.35±3.14°; F[1, 37]=7.00, p=0.012, $\eta$ 2=0.15), suggesting that the range of spatial disparities at which ASD participants perceived a common source was significantly smaller than in controls. Note, this range is well beyond the 6° used in the maximum likelihood estimation experiment (~fourfold), thus corroborating that during the first experiment participants perceived auditory and visual cues as a single, multisensory cue.

To further substantiate these differences in the explicit report of common cause across ASD and neurotypical subjects, we next dissociated auditory and visual cues across time, as opposed to space. Twenty-one individuals with ASD (15.94±0.56 years; 5 females) and 13 age-matched neurotypical controls (16.3±0.47 years; 5 females, see *Supplementary file 1*, *Supplementary file 2*) viewed a visual disk and heard an auditory tone, either in synchrony (Δ=0 ms) or over a wide range of asynchronies (from ±10 to ±700 ms; positive Δ indicates visual led auditory stimulus). Subjects indicated if auditory and visual stimuli occurred synchronously or asynchronously.

Analogous to the case of spatial disparities, we fit reports of common cause (i.e. synchrony, in this case) to Gaussian functions. Just as for spatial disparities, the ASD group had smaller amplitudes (ASD = 0.83±0.04; control = 0.98±0.01; *Figure 3D*; t-test: $t_{32}$=7.75, p<0.001, Cohen's d>2), suggesting that at small Δ individuals with ASD perceived the stimuli as originating from a common cause less frequently than control subjects did. Further, the ASD group exhibited larger Gaussian widths (control = 171.68±13.17; ASD = 363±55.63 ms; t-test: $t_{32}$=2.61, p=0.01, Cohen's d=0.9), reflecting more frequent reports of common cause at large temporal disparities. This second effect corroborates a multitude of reports demonstrating larger 'temporal binding windows' in ASD than control (see *Feldman et al., 2018* for a meta-analysis of 53 studies). Overall, therefore, explicit reports of common cause across spatial and temporal disparities agree in suggesting a lower likelihood of inferring a common cause at small temporal disparities - including no disparity - in ASD relative to neurotypical controls (see e.g. *Noel et al., 2018b*; *Noel et al., 2018a*, for previous reports showing altered overall tendency to report common cause during temporal disparities in ASD, although these reports typically focus on the size of 'binding windows').

Correlational analyses between psychometric features distinguishing control and ASD individuals (i.e. linear and cubic terms accounting for auditory biases during large audio-visual spatial disparities, amplitude and width of explicit common cause reports during spatial and temporal disparities) and symptomatology measures, i.e., autism quotient (AQ; *Baron-Cohen et al., 2001*) and social communication questionnaire (SCQ; *Rutter et al., 2003*) demonstrated weak to no association. Of the 12 correlations attempted ([AQ + SCQ] × [amplitude + width] × [temporal + spatial] + [AQ + SCQ] × [linear + cubic terms]), the only significant relation (surviving Bonferroni-correction) was that between

the width of the Gaussian function describing synchrony judgments as a function of temporal disparity and SCQ scores (Type II regression: $r$=0.52, p=0.002; see *Smith et al., 2017* for a similar observation).

## Causal inference modeling suggests an increased prior probability for common cause in ASD

To bridge across experiments (i.e. implicit and explicit audio-visual spatial tasks) and provide a quantitative account of the switch between internal models (i.e. segregate vs. integrate) in ASD vs. controls, we fit subjects' responses with a Bayesian causal inference model (*Figure 4A* and *Körding et al., 2007*). The modeling effort is split in three steps.

First, we fit aggregate data and attempt to discern which of the parameters that govern the causal inference process may globally differ between the ASD and control cohorts. The parameters of the causal inference model can be divided into three sets. First, sensory parameters: the visual and auditory sensory uncertainty (i.e. inverse of reliability), as well as visual and auditory priors (i.e. expectations) over the perceived auditory and visual locations (mean and variance of Gaussian priors). Second, choice parameters: choice bias ($p_{choice}$), as well as lapse rate and bias. These latter two parameters are the frequency with which an observer may make a choice independent of the sensory evidence (lapse rate) and whether these stimuli-independent judgments are biased (lapse bias). Third, inference parameters: the prior probability of combination ($p_{common}$; see *Methods and Supplementary file 3*, *Supplementary file 4* for further detail). In this first modeling step, we fit all parameters (see *Supplementary file 3*) to best account for the aggregate control subject. Then, we test whether a difference in choice and inference parameters, but not the sensory ones, can explain the observed difference between the control and the aggregate ASD data. We do not vary the sensory parameters given that unisensory discrimination thresholds did not differ between experimental groups (*Figure 1*, *Figure 2—figure supplement 1*, and *Figure 3—figure supplement 1*. See *Methods*, *Supplementary file 4* and *Figure 4—figure supplement 1* for technical detail regarding the model fitting procedure. Also see *Figure 4—figure supplement 2* corroborating the fact that varying the inference parameter, as opposed to sensory uncertainty, results in better model fits). In a second step, we attempt not to globally differentiate between ASD and control cohorts, but to account for individual subject behavior. Thus, we fit single subject data and utilize the subject-specific measured sensory uncertainty to fit all parameters (i.e. sensory, choice, and inference). All subjects who completed the cue integration experiment (Experiment 1) – allowing for deriving auditory and visual localization thresholds – and either the implicit (Experiment 2) or explicit (Experiment 3) spatial causal inference task were included in this effort. This included 'poor performers' (six in Experiment 1 and eight in Experiment 3), given that the goal of this second modeling step was to account for individual subject behavior. Last, we perform model comparison between the causal inference model and a set of alternative accounts, also putatively differentiating the two experimental groups.

*Figure 4B and C*, respectively, shows the aggregate control and ASD data for the implicit and explicit causal inference task (with each panel showing different visual reliabilities). In the implicit task (*Figure 4B*, top panel), allowing only for a difference in the choice parameters (lapse rate, bias, and $p_{choice}$; magenta) between the control and ASD cohorts, could only partially account for observed differences between these groups (explainable variance explained, EVE=0.91, see *Supplementary file 4*). Instead, differences between the control and ASD data could be better explained if the prior probability of combining cues, $p_{common}$, was also significantly higher for ASD relative to control observers (*Figure 4D*, p=4.5 × 10$^{-7}$, EVE=0.97, ΔAIC between model varying only choice parameters vs. choice and inference parameters = 1 × 10$_{3}$). This suggests the necessity to include $p_{common}$ as a factor globally differentiating between the neurotypical and ASD cohort.

For the explicit task, different lapse rates and biases between ASD and controls could also not explain their differing reports (as for the implicit task; EVE = 0.17). Differently from the implicit task, however, we cannot dissociate the prior probability of combination (i.e. $p_{common}$) and choice biases, given that the report is on common cause (*Figure 4A*, see *Methods* and *Supplementary file 4* for additional detail). Thus, we call the joint choice and inference parameter $p_{combined}$ (this one being a joint $p_{common}$ and $p_{choice}$). Allowing for a lower $p_{combined}$ in ASD could better explain the observed differences between ASD and control explicit reports (*Figure 4C*; EVE = 0.69, ΔAIC relative to a model solely varying lapse rate and bias = 1.3 × 10$^{3}$). This is illustrated for the ASD aggregate subject relative to the aggregate control subject in *Figure 4D* (p=1.8 × 10$^{-4}$). Under the assumption that an observer's

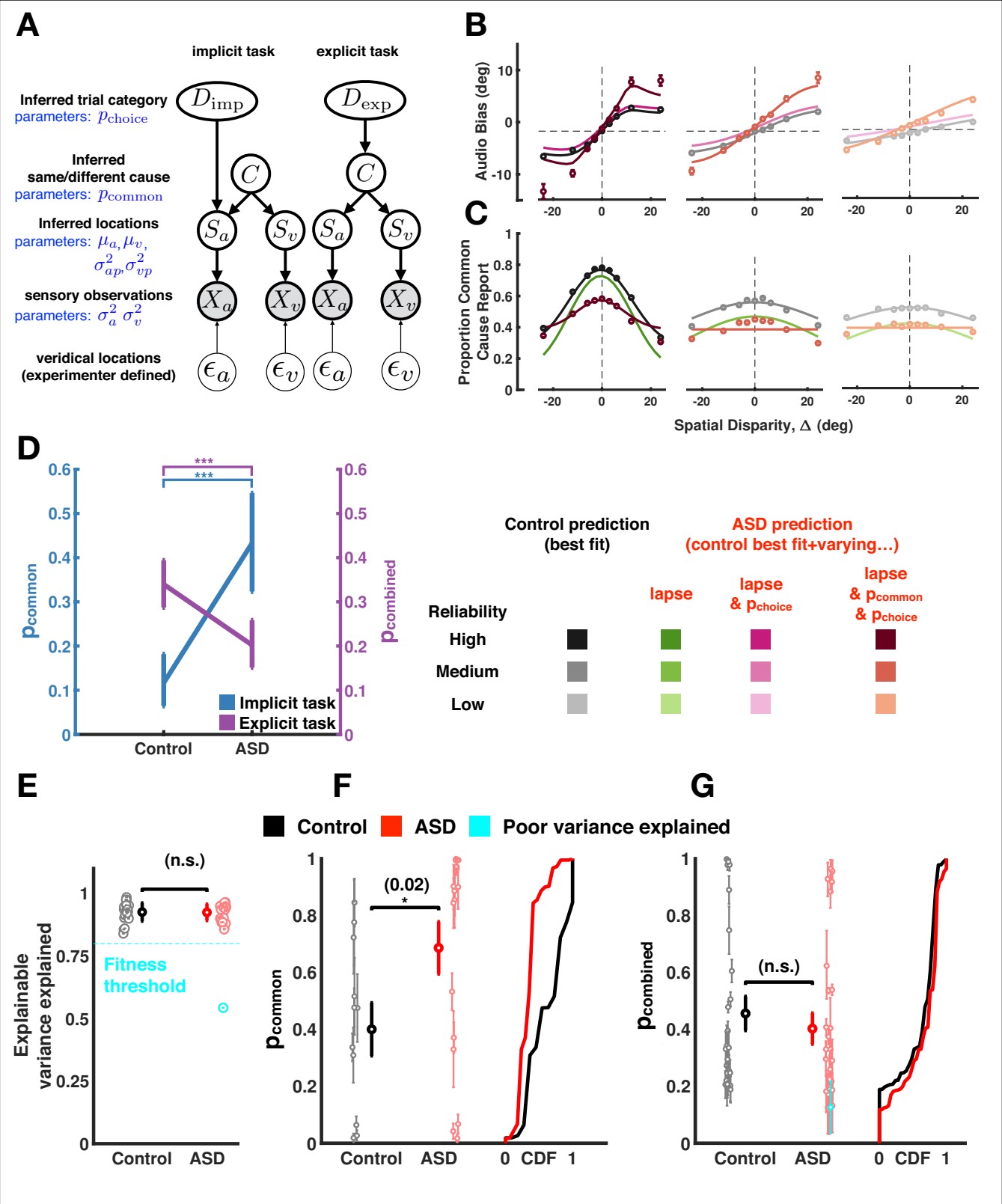

**Figure 4.** Causal inference modeling of implicit and explicit spatial tasks. (**A**) Generative models of the causal inference process in the two tasks (implicit task in left and explicit task in right). The subject makes noisy sensory measurements (*X*) of the veridical cue locations (*ε* and combines them with their prior belief to obtain their percept (*S*). To do so optimally, the subject first must infer whether signals came from the same cause (*C*) and thereby determine if it is useful to combine the information from the two cues for inferring the trial category (*D*). The causal inference process is shared

*Figure 4 continued on next page*

*Figure 4 continued*

between the two tasks but the subject infers $D_{imp}$ (side of the tone) in the implicit task and $D_{exp}$ (number of causes for the sensory observations) in the explicit task. (**B**) Aggregate data (dots) and model fits (lines) in the implicit task (the visual reliability varies from high to low from left to right). The causal inference model is fit to the control aggregate subject and different set of parameters are varied to match the autism spectrum disorder (ASD) subject data (see main text). See *Figure 4—figure supplement 12* for a fit to the same data while (1) allowing all parameters free to vary, (2) allowing the same parameter as here to vary, but fitting to visual reliabilities separately, or (3) doing both (1) and (2). Of course, these result in better fits, but this is at the expense of interpretability in that they are inconsistent with the empirical data. (**C**) Same as (**B**) but fits are to the explicit spatial task. See *Figure 4—figure supplement 13* for the equivalent of *Figure 4—figure supplement 12*, for the implicit task. Data (dots) are slightly different from that in *Figures 2 and 3* because in the previous figures data was first averaged within subjects, then psychometric functions were fit, and finally estimates of bias were averaged across subjects. Here, data is first aggregated across all subjects and then psychometric fits are done on the aggregate. Importantly, the difference between ASD and control subjects holds either way. Error bars are 68% CI (see *Supplementary file 4* for additional detail regarding deriving CIs for the amalgamated subject). (**D**). ASD subjects have a higher p-common for the aggregate subject in the implicit task but seemingly compensate in the explicit task where they show a lower aggregate p-common and choice bias. (**E**). The causal inference model provides an equally good fit (quantified by explainable variance explained), a measure of goodness of fit appropriate for noisy, as opposed to noiseless data (*Haefner and Cumming, 2008*) for control and ASD subjects. (**F**) Individual ASD (red) subjects have a higher p-common on average for the implicit task (in agreement with the aggregate subject) but (**G**) show no significant difference in the combined p-common and choice bias for the explicit task due to considerable heterogeneity across subjects. Subjects were included in the single-subject modeling effort if they had participated in Experiment 1 (and thus we had an estimate of their sensory encoding) in addition to the particular task of interest. That is, for panel (**F**), we included all participants taking part in Experiments 1 and 2. This included participants deemed poor in Experiment 1, given our attempt to account for participant's behavior with the causal inference model. For panel (**G**), we included all participants taking part in Experiments 1 and 3. Individual subject error bars are 68% CI, while group-level error bars are 95% CI (see *Supplementary file 4* for additional detail regarding statistical testing). CDF = cumulative density function.

The online version of this article includes the following figure supplement(s) for figure 4:

**Figure supplement 1.** Flowchart detailing steps in fitting procedure.

**Figure supplement 2.** Fit to aggregate data for the implicit causal inference task, allowing sensory uncertainty and choice parameters to vary but fixing the inference parameter $p_{common}$ (shown in pink).

**Figure supplement 3.** Data from a single, representative control subject.

**Figure supplement 4.** Data from another single, representative control subject.

**Figure supplement 5.** Data from a single, representative autism spectrum disorder (ASD) subject.

**Figure supplement 6.** Data from another single, representative autism spectrum disorder subject.

**Figure supplement 7.** Goodness of fit of alternative models for the implicit and explicit spatial causal inference task.

**Figure supplement 8.** Illustration of the alternative models fits to implicit causal inference model data.

**Figure supplement 9.** Illustration of the alternative models fits to explicit causal inference model data.

**Figure supplement 10.** Causal inference modeling of temporal, simultaneity judgment task.

**Figure supplement 11.** Lapse rate and lapse bias for aggregate and individual subjects during the implicit and explicit spatial tasks.

**Figure supplement 12.** Fit to aggregate data for the implicit causal inference task, given that all parameters are free to vary (**A**), the different visual reliabilities are fit separately (**B**) or both of the above (**C**).

**Figure supplement 13.** Fit to aggregate data for the explicit causal inference task, given that all parameters are free to vary (**A**), the different visual reliabilities are fit separately (**B**) or both of the above (**C**).

expectation for cues to come from the same cause ($p_{common}$) is formed over a long timescale, and hence is the same across the implicit and explicit tasks, we can ascribe the differing pattern of results in the tasks (i.e. increased $p_{common}$ in ASD in the implicit task, yet a decreased $p_{combined}$ in the explicit task) to differences in the choice bias (i.e. the added component from $p_{common}$ to $p_{combined}$). This bias may in fact reflect a compensatory strategy by ASD observers since we found their $p_{common}$ (uncorrupted by explicit choice biases) to be roughly three times as large as that of the aggregate control observer (*Figure 4D*).

Next, we fit the model to individual subject data (as opposed to the aggregate) and obtained full posterior estimates over all model parameters for individual observers. We fit the model jointly to unisensory and causal inference tasks, such that we can constrain the sensory parameters by the observed unisensory data (*Figure 1*). The causal inference model provided a good and comparable fit for both ASD and control subjects (*Figure 4E*) with the model explaining more than 80% of explainable variance in all but one subject (*Figure 4E*, blue dot). *Figure 4—figure supplements 3–6* show individual data for two representative control (*Figure 4—figure supplements 3 and 4*) and two ASD

subjects (*Figure 4—figure supplements 5 and 6*), while highlighting all the data that constrained the model fits (audio localization, visual localization at three reliabilities, forced fusion task at three reliabilities, as well as implicit and explicit causal inference). Overall, both groups were heterogeneous (*Figure 4F and G*). Nonetheless, in agreement with the aggregate data, individuals with ASD had a higher prior probability of common cause than control subjects (*Figure 4F*) during the implicit task (p=0.02), where $p_{common}$ can be estimated independently from $p_{choice}$. When estimating $p_{combined}$ (i.e. the combination of $p_{common}$ and $p_{choice}$) for the explicit task (*Figure 4G*), the parameter estimates extracted from the individual fits suggested no difference between ASD and control subjects (p=0.26), although numerically the results are in line with the aggregate data, suggesting a lower $p_{combined}$ in ASD than control (see inter-subject variability in *Figure 4F and G*). Importantly, the aggregate and single subject fits concord in suggesting an explicit compensatory mechanism in individuals with ASD, given that $p_{common}$ is higher in ASD than control (when this parameter can be estimated in isolation) and a measure corrupted by explicit choice biases (i.e. $p_{combined}$) is not. Individual subjects' $p_{common}$ and $p_{combined}$ as estimated by the model did not correlate with ASD symptomatology, as measured by the AQ and SCQ (all p>0.17). Exploration of the model parameters in the 'poor performers' did not suggest a systematic difference between these subjects and other vis-à-vis their causal inference parameters.

Last, we consider a set of alternative models that could in principle account for differences in behavior across the aggregate control and ASD cohorts. The first alternative (alternative A) was a forced fusion model where all parameters were fit to the ASD aggregate subject, but $p_{common}$ was fixed to a value of 1. Thus, under this account the ASD subject always combines the cues irrespective of the disparity between them. Alternative B was a no fusion model, the opposite to Alternative A, where now all parameters were fit to the ASD aggregate subject, but $p_{common}$ was fixed to a value of 2. Alternative C had a lapse rate but no lapse bias. Last, alternative D allowed only the choice parameters to vary between control and ASD, but no inference or sensory parameter. For the implicit task, lapse rate, bias, and $p_{choice}$ were allowed to vary. For the explicit task since $p_{choice}$ trades off with $p_{common}$, only lapse rate and bias were allowed to vary.

We performed model comparison using AIC and *Figure 4—figure supplement 7* shows this metric for the ASD aggregate subject relative to the causal inference model where we vary choice and inference parameters (i.e. the model used in *Figure 4*. Lower AIC indicates a better fit). *Figure 4—figure supplement 8* and *Figure 4—figure supplement 9* show the original (choice and inference) and alternative fits, respectively, to implicit and explicit spatial causal inference tasks. For the implicit task, varying sensory and choice parameters, as opposed to inference parameters, results in a worse quality fit. Interestingly, alternative A (forced fusion) is a considerably better model than alternative B (forced segregation). Together, this pattern of results suggests that choice and inference (and not choice and sensory) parameters distinguish between ASD and control subjects in the implicit causal inference task. Likewise, these results further corroborate the conclusion that ASD subjects favor an internal model where integration outweighs segregation (AIC alternative A<AIC alternative B), yet there is not a complete lack of causal inference in ASD, given that alternative A is inferior to the model where $p_{common}$ is less than 1. In other words, individuals with ASD do perform causal inference, but they give more weight to integration (vs. segregation) compared to neurotypical subjects. For the explicit task, the alternative models considered performed worse than allowing the choice and inference parameters to vary (main model used in *Figure 4*).

For completeness, we fit the causal inference model to data from the simultaneity judgment task (see *Figure 4—figure supplement 10* and *Supplementary file 5*), given that this task constitutes a large portion of the literature on multisensory impairments in ASD (see e.g. *Feldman et al., 2018*). However, in this task, given its explicit nature, it is also not possible to dissociate $p_{choice}$ and $p_{common}$ (as for the explicit spatial task), and even more vexingly, given that reliabilities were not manipulated (as is typical in the study of multisensory temporal acuity, see *Nidiffer et al., 2016*, for an exception), it is also difficult to dissociate the $p_{choice}$ from lapse parameters with a reasonable amount of data. We also explore the impact of lapse rates and biases and their differences across ASD and control subjects in *Figure 4—figure supplement 11*.

## Discussion

We presented individuals with ASD and neurotypical controls with audio-visual stimuli at different spatial or temporal disparities, and measured their unisensory spatial discrimination thresholds, their

implicit ability to perform optimal cue combination, and their implicit and explicit tendency to deduce different causal structures across cue disparities. The results indicate no overall impairment in the ability to perform optimal multisensory cue integration (*Ernst and Banks, 2002*). These observations generalize a previous report (*Zaidel et al., 2015*) and suggest that across domains (visuo-vestibular in *Zaidel et al., 2015* audio-visual here), optimal cue combination is intact in ASD. Instead, we found that even at large spatial disparities, individuals with ASD use information from one sensory modality in localizing another. That is, in contrast to neurotypical controls, individuals with ASD behaved as if they were more likely to infer that cues come from the same rather the different sources. This suggests that the well-established anomalies in multisensory behavior in ASD - e.g., biases (see *Baum et al., 2015* and *Wallace et al., 2020*, for reviews) – may not be due to a dysfunctional process of multisensory integration per se, but one of impair causal inference.

The juxtaposition between an impaired ability for causal inference yet the presence of an intact ability for optimal cue combination may suggest a deficit in a specific kind of computation and point toward anomalies in particular kinds of neural motifs. Indeed, an additional algorithmic component in causal inference (*Körding et al., 2007*) relative to optimal cue combination models (*Ernst and Banks, 2002*) is the presence of non-linear operations such as marginalization. This operation corresponds to 'summing out' nuisance variables, allows for non-linearities, and may be neurally implemented via divisive normalization (see *Beck et al., 2011* for detail on marginalization and the relationship between this operation and divisive normalization). In fact, while not all proposed neural network models of causal inference rely on divisive normalization (see *Cuppini et al., 2017*; *Zhang et al., 2019* for networks performing causal inference without explicit marginalization), many do (e.g. *Yamashita et al., 2013*; *Yu et al., 2016*). Divisive normalization is a canonical neural motif (*Carandini and Heeger, 2011*), i.e., thought to operate throughout the brain, wherein neural activity from a given unit is normalized by the joint output of a normalization neural pool. Thus, the broad anomalies observed in ASD may be underpinned by an alteration in a canonical computation, i.e., causal inference, which in turn is dependent on a canonical neural motif, i.e., divisive normalization. *Rosenberg et al., 2015*, suggested that anomalies in divisive normalization – specifically a reduction in the amount of inhibition that occurs through divisive normalization – —can account for a host of perceptual anomalies in ASD, such as altered local vs. global processing (*Happé and Frith, 2006*), altered visuo-spatial suppression (*Foss-Feig et al., 2013*), and increased tunnel vision (*Robertson et al., 2013*). This suggestion – from altered divisive normalization, to altered marginalization, and in turn altered causal inference and multisensory behavior – is well aligned with known physiology in ASD and ASD animal models showing decrease GABAergic signaling (*Lee et al., 2017*; *Chen et al., 2020*), the comorbidity between ASD and seizure activity (*Jeste and Tuchman, 2015*), and the hypothesis that ASD is rooted in an increased excitation-to-inhibition ratio (i.e. E/I imbalance; *Rubenstein and Merzenich, 2003*).

A second major empirical finding is that individuals with ASD seem to explicitly report common cause less frequently than neurotypical controls. Here we demonstrate a reduced tendency to explicitly report common cause during small cue disparities, across both spatial and temporal disparities (also see *Figure 2—figure supplement 2E-G* for corroborative evidence during a motion processing task). This has previously been observed within the temporal domain (*Noel et al., 2018b*; *Noel et al., 2018a*), yet frequently multisensory simultaneity judgments are normalized to peak at '1' (e.g. *Woynaroski et al., 2013*; *Dunham et al., 2020*), obfuscating this effect. To the best of our knowledge, the reduced tendency to explicitly report common cause across spatial disparities in ASD has not been previously reported. Further, it is interesting to note that while 'temporal binding windows' were larger in ASD than control (see *Feldman et al., 2018*), 'spatial binding windows' were smaller in ASD relative to control subjects. This pattern of results highlights that when studying explicit 'binding windows', it may not be sufficient to index temporal or spatial domains independently, but there could potentially be a trade-off. More importantly, the reduced tendency to overtly report common cause across spatial and temporal domains in ASD (even when implicitly they seem to integrate more, and not less often) is indicative of a choice bias that may have emerged as a compensatory mechanism to their increased implicit tendency to bind information across sensory modalities. This speculation is supported by formal model fitting, where the prior probability of combination (p-common) was larger at the (aggregate) population level in the ASD than the control subjects in implicit tasks (where p-common may be independently estimated), yet a combined measure of p-common and a choice bias (these not being dissociable in explicit tasks such as spatial or temporal common cause

reports) that was reduced (in the aggregate) or not significantly different (in the individual subject data) between ASD and control individuals. The presence of this putative compensatory mechanism is important to note, particularly when a significant fraction of the characterization of (multi)sensory processing in ASD relies on explicit tasks. Further, this finding, highlights the importance in characterizing both implicit and explicit perceptual mechanisms – particularly when framed under a strong theoretical foundation (*Ernst and Banks, 2002*; *Körding et al., 2007*) and using model-based analyses (e.g. *Lawson et al., 2017*; *Lieder et al., 2019*) – given that explicit reports may not faithfully reflect subjects' percepts.

Last, it is also interesting to speculate on how an increased prior probability of integrating cues, and the presence of a compensatory mechanism, may relate to ASD symptomatology. Here we did not observe any reliable correlation between symptomatology and either psychophysical measures or model parameter estimates. However, it must be acknowledged that while the overall number of participants across all experiments was relatively large (91 subjects in total), our sample sizes within each experiment were moderate (~20 subjects per group and experiment), perhaps explaining the lack of any correlation. Regardless, it is well established that beyond (multi)sensory anomalies (*Baum et al., 2015*), individuals with ASD show inflexible and repetitive behaviors (*Geurts et al., 2009*) and demonstrate 'stereotypy', self-stimulatory behaviors thought to relieve sensory-driven anxiety (*Cunningham and Schreibman, 2008*). The finding that individuals with ASD do not change their worldview (i.e. from integration to segregation, even at large sensory disparities) may result in sensory anomalies and reflect the slow updating of expectations (*Vishne et al., 2021*). Thus, anomalies in causal inference may have the potential of explaining seemingly disparate phenotypes in ASD – anomalous perception and repetitive behaviors. Similarly, we may conjecture that stereotypy is a physical manifestation of a compensatory mechanism, such as the one uncovered here. Stereotypy could result from attempting to align incoming sensory evidence with the (inflexible) expectations of what that sensory input ought to be.

In conclusion, by leveraging a computational framework (optimal cue combination and causal inference; *Ernst and Banks, 2002*; *Körding et al., 2007*) and systematically measuring perception at each step (i.e. unisensory, forced cue integration, and causal inference) across a range of audio-visual multisensory behaviors, we can ascribe anomalies in multisensory behavior to the process of inferring the causal structure linking sensory observations to their hidden causes. Of course, this anomaly results in perceptual biases (see the current results and *Baum et al., 2015* for an extensive review), but the point is that these biases are driven by a canonical computation that has gone awry. Further, given the known E/I imbalance in ASD (*Rubenstein and Merzenich, 2003*; *Lee et al., 2017*; *Chen et al., 2020*) and the fact that causal inference may require marginalization but optimal cue combination does not (*Beck et al., 2011*), we can speculatively suggest a bridge from neural instantiation to behavioral computation; E/I imbalance may disrupt divisive normalization (neural implementation), which leads to improper marginalization (algorithm) and thus altered causal inference (computation) and multisensory perception (biases in behavior) in ASD.

# Materials and methods
## Participants
A total of 91 adolescents (16.25±0.4 years; 20 females) took part (completely or partially) in a series of up to five behavioral experiments (four audio-visual and presented in the main text, in addition to a visual heading discrimination task presented in the *Supplementary Materials*). Forty-eight of these were neurotypical controls. Individuals in the control group (16.5±0.4 years; 13 females) had no diagnosis of ASD or any other developmental disorder or related medical diagnosis. These subjects were recruited by flyers posted throughout Houston. The other 43 participants (16.0±0.5 years; 7 females) were diagnosed as within ASD. The participants with ASD were recruited through several sources, including (1) the Simons Simplex Collection families, (2) flyers posted at Texas Children's Hospital, (3) the Houston Autism Center, and (4) the clinical databases maintained by the Simons Foundation Autism Research Initiative (SFARI). All participants were screened at enrollment with SCQ (*Rutter et al., 2003*) and/or the AQ (*Baron-Cohen et al., 2001*) to afford (1) a measure of current ASD symptomatology and (2) rule out concerns for ASD in control subjects. There was no individual with ASD below the recommended SCQ cutoff, and only 2 (out of 47) control subjects above this cutoff (*Surén*

*et al., 2019*). Similarly, there was almost no overlap in ASD and control AQ scores (with only 3 out of 47 control individuals having a higher AQ score than the lowest of the individuals with ASD). All individuals with ASD were above the AQ cutoffs recommended by *Woodbury-Smith et al., 2005* and *Lepage et al., 2009* (respectively, cutoff scores of 22 and 26), but not by *Baron-Cohen et al., 2001* (cutoff score of 36). Inclusion in the ASD group required that subjects have (1) a confirmed diagnosis of ASD according to the DSM-5 (*American Psychiatric Association, 2013*) by part of a research-reliable clinical practitioner and (2) no history of seizure or other neurological disorders. A subset of the individuals with ASD were assessed by the Autism Diagnostic Observation Schedule (ADOS-2, *Lord et al., 2012*), and no difference was observed in the AQ, SCQ, or psychometric estimates between individuals with ASD with and without the ADOS assessment (all p>0.21). Similarly, the intelligence quotient (IQ) as estimated by the Wechsler Adult Intelligence Scale (WAIS) was available for a subset of the ASD participants (n=10, or 22% of the cohort), whose average score was 103±9 (S.E.M.), this being no different from the general population (which by definition has a mean of 100). All subjects had normal visual and auditory acuity, as characterized by parents' and/or participants' reports. For each of the five psychophysics experiments, we aimed at scheduling approximately 25–30 participants per group, in accord with sample sizes from previous similar reports (*Dokka et al., 2019*; *Noel et al., 2018b*). Data were not examined until after data collection was complete. The study was approved by the Institutional Review Board at the Baylor College of Medicine (protocol number H-29411) and written consent/assent was obtained.

## Experimental materials and procedures

### Experiment 1: Audio-visual spatial localization; maximum-likelihood estimation (implicit)

Thirty-one ASD (age = 15.2±0.4 years) and 34 control (16.1±0.4 years) subjects participated in this experiment. As expected, the SCQ (ASD = 17.1±0.75; control = 4.8±0.5; t-test: $t_{63}$=−13.31, p<0.0001) and AQ scores (ASD = 31.2±1.7; control = 15.3±1.5; $t_{41}$=−6.61, p<0.0001) of the ASD group were significantly greater than that of the control group.

Subjects performed a spatial localization task of auditory, visual, or combined audio-visual stimuli. A custom-built setup comprising of (1) an array of speakers and (2) a video projection system delivered the auditory and visual stimuli, respectively. Seven speakers (TB-F Series; W2-852SH) spaced 3° apart were mounted on a wooden frame along a horizontal line. A video projector (Dell 2,400 MP) displayed images onto a black projection screen (60 × 35°) that was mounted over the speaker array. This arrangement allowed presentation of the visual stimulus precisely at the location of the auditory stimulus, or at different locations on the screen. The auditory stimulus was a simple tone at 1200 Hz. The visual stimulus was a white circular patch. Reliability of the visual stimulus was manipulated by varying the size of the visual patch such that reliability inversely varied with the patch size (*Alais and Burr, 2004*). Three levels of visual reliability were tested: high (higher reliability of visual vs. auditory localization), medium (similar reliabilities of visual and auditory localization), and low (poorer reliability of visual vs. auditory localization). For high and low visual reliabilities, the patch diameter was fixed for all participants at 5 and 30°, respectively. For medium reliability, the patch diameter ranged from 15 to 25° across subjects. In all conditions (audio-only, visual-only, or combined audio-visual), the auditory and/or visual stimuli were presented for 50 ms (and synchronously in the case of combined stimuli). Stimuli were generated by custom MATLAB scripts employing the PsychToolBox (*Kleiner et al., 2007*; *Noel et al., 2022*).

Subjects were seated 1 m from the speaker-array with their chins supported on a chinrest and fixated a central cross. Subjects performed a single-interval, two-alternative-forced-choice spatial localization task. In each trial, they were presented with either an auditory, visual, or combined audio-visual stimulus (*Figure 1A*). They indicated if the auditory and/or visual stimulus were located to the left or right of straight forward by button-press. The spatial locations of the stimuli were varied in steps around straight forward. In single-cue auditory and combined conditions, the auditory stimulus was presented at one of the seven locations: 0,±3,±6, and ±9° (positive sign indicates that the stimulus was presented to the right of the participant). By contrast, the visual stimulus could be presented at any location on the screen. Specifically, in the single-cue visual condition, the visual stimulus was presented at ±20, ±10, ±5, ±2.5, ±1.25, ±0.65, ±0.32, and 0°. In the combined condition, auditory and visual stimuli were either presented at the same spatial location (*Figure 1*, top panel; Δ=0°) or

at different locations separated by a spatial disparity Δ=±6° (*Figure 1A*, bottom panel; positive Δ indicates that the auditory stimulus was located to the right of the visual stimulus). For trials in which there was a spatial conflict, a mean stimulus location was defined. The auditory and visual stimuli were presented on either side of this mean stimulus location at an angular distance of Δ/2. For Δ=6°, the mean stimulus was located at –12, –9, –6, –3, 0, 3, and 6°. For Δ=–6°, the mean stimulus was located at –6, –3, 0, 3, 6, 9, and 12°. Each subject performed a total of 1680 trials (auditory condition = 7 stimulus locations × 15 repetitions; visual condition = 14 stimulus locations × 15 repetitions × 3 visual cue reliabilities; and combined auditory-visual condition = 7 stimulus locations × 3 reliabilities × 3 conflict angles × 15 repetitions). All conditions were interleaved.

For each subject, visual cue reliability, stimulus condition, and spatial disparity, psychometric functions were constructed by plotting the proportion of rightward responses as a function of stimulus location. These data were fit with a cumulative Gaussian function using *psignifit*, a MATLAB package that implements the maximum-likelihood method (*Wichmann and Hill, 2001*). The psychometric function yields two parameters that characterize participants' localization performance: bias and threshold. Bias (μ) is the stimulus value at which responses are equally split between rightward and leftward. A bias close to 0° indicates highly accurate localization. The threshold is given by the SD (σ) of the fitted cumulative Gaussian function. The smaller the threshold, the greater the precision of spatial localization. The bias and threshold values estimated from these psychometric functions were used to test the predictions of optimal cue integration. The psychometric fitting could not estimate auditory thresholds for six ASD subjects, whose report did not vary as a function of auditory stimuli location. These subjects were not included in the remaining analyses reported in the main text.

Based on unisensory localization, we may derive predictions for the combined case, given optimal cue combination by maximum-likelihood estimation (*Ernst and Banks, 2002*; *Hillis et al., 2002*; *Alais and Burr, 2004*; *Kersten et al., 2004*). First, assuming optimal cue combination, the threshold in the combined auditory-visual condition ($\sigma_{com}$) should be equal to:

$$\sigma_{com} = \sqrt{\frac{\sigma_a^2 \sigma_v^2}{\sigma_a^2 + \sigma_v^2}} \qquad (1)$$

with $\sigma_a$ and $\sigma_v$ being the thresholds in the unisensory auditory and visual localization, respectively. Second, the weight assigned to the visual cue in combined audio-visual stimuli (see *Ernst and Banks, 2002* and *Alais and Burr, 2004*, for detail) should vary with its reliability. Specifically, as visual cue reliability decreases, the visual weight will also decrease. The visual weight, $w_v$, is predicted to be:

$$W_v = \frac{\frac{1}{\sigma_v^2}}{\frac{1}{\sigma_v^2} + \frac{1}{\sigma_a^2}} \qquad (2)$$

and in turn the auditory cue weight ($w_a$) is computed as $1 - w_v$.

## Experiment 2: Audio spatial localization with disparate visual cues; causal inference (implicit)

Twenty-two ASD (age = 17.32±0.57 years) and 15 control (age = 16.86±0.55 years) subjects participated in this experiment. As expected, the SCQ (ASD = 16.42±1.12; control = 5.06±0.65; t-test: $t_{35}$=7.84, p<0.0001) and AQ scores (ASD = 31.95±1.76; control = 13.76±1.61; $t_{35}$=7.21, p<0.0001) of the ASD group were significantly greater than that of the control group.

The task and stimuli employed here were identical to the audio-visual localization experiment described above, except that a larger range of spatial disparities were employed. The disparity between cues (Δ) could take one of nine values: 0, ±3, ±6, ±12, and ±24°. Each Δ was presented 8 times at each of the 7 speaker locations, and at each visual cue reliability, resulting in a total of 1512 trials (9 spatial disparities × 7 speaker locations × 3 reliabilities × 8 repetitions). Subjects indicated if the auditory stimulus was located to the right or left of straight ahead. Subjects were informed that the flash and beep could appear at different physical locations. All conditions were interleaved, and subjects were required to take breaks and rest after each block.

For each subject, audio-visual disparity (Δ), and visual cue reliability, psychometric functions were constructed by plotting the proportion of rightward responses as a function of the true auditory stimulus location. As for the audio-visual localization task described above, data were fitted with

a cumulative Gaussian function. An auditory bias close to 0° indicates that the subject was able to discount the distracting influence of the visual cues and accurately localize the audio beep. Data from one ASD subject was excluded from this analysis as the subject was unable to perform the task even when auditory and visual stimuli were co-localized (Δ=0°). In eight ASD subjects, psychometric functions could not fit into the data even at the highest disparity (Δ = ±24°) during high reliability, as subjects' estimates were 'captured' by the visual cues. The remaining data from these subjects were included in the analyses.

As an initial quantification of localization estimates, and putative differences in audio-visual biases between the groups, a third-order regression model of the form: $y = a_0 + a_1\Delta + a_2\Delta^2 + a_3\Delta$ *American Psychiatric Association, 2013* was fitted to the auditory bias as a function of Δ and visual cue reliability. Coefficient $a_1$ represents how sensitive the bias is to changes in Δ - larger $a_1$ indicates a greater change in the bias for a given change in Δ. Coefficient $a_2$ indicates if the dependence of bias on Δ is uniform for positive and negative Δ values. Importantly, coefficient $a_3$ generally represents how the bias changes at large Δ values – negative $a_3$ indicates a saturation or a decrease in the bias at large Δ. If subjects perform causal inference (*Körding et al., 2007*), we expect a saturation or even a return to no bias at large Δ. Furthermore, partial $R^2$ values associated with $a_1$, $a_2$, and $a_3$ describe the contribution of each term in explaining the total variance. ASD and control subjects' data was well-explained by the third-order regression model (ASD: $R^2=0.93\pm0.04$; control: $R^2=0.88\pm0.03$). A mixed-effects ANOVA with group, Δ, and visual cue reliability as factors compared the bias, threshold, and parameters of the regression model for the ASD and control groups.

## Experiment 3: Audio-visual common source reports under spatial disparities (Explicit)

Twenty-three23 ASD (age = 16.14±0.51 years) and 24 control (age = 17.10±0.42 years) subjects participated in this experiment. Six other ASD subjects were screened for this experiment, but showed poor auditory localization (c.f. Experiment 1). The SCQ (ASD = 16.91±0.83; control = 5.04±0.47; t-test: $t_{57}=11.46$, $p<0.0001$) and AQ scores (ASD = 30.77±1.60; control = 15.18±1.60; $t_{41}=6.42$, $p<0.0001$) of the ASD group were significantly greater than that of the control group.

The auditory and visual stimuli presented in this task were identical to those employed in Experiment 2. Each Δ was presented 7 times, at each of seven speaker locations, and at each visual cue reliability, resulting in a total of 1323 trials (9 spatial disparities × 7 speaker locations × 3 reliabilities × 7 repetitions). Subjects indicated via button-press if the auditory and visual cues originated from a common source or from different sources. The exact instructions were to "press the 'same source' key if auditory and visual signals come from the same source, and press the 'different sources' key if auditory and visual signals come from different sources." All conditions were interleaved, and subjects were required to take breaks and rest after each block. Before the start of the main experiment, subjects participated in a practice block to familiarize themselves with the stimuli and response buttons. The response buttons (one for 'same source' and the other for 'different sources' were the left and right buttons of a standard computer mouse. Reports from eight ASD subjects did not vary with Δ, and thus their data was excluded from the main analyses).

For each subject, audio-visual disparity (Δ), and visual cue reliability, the proportion of common source reports was calculated. A mixed-effects ANOVA with group as the between-subjects factor, along with Δ and visual cue reliability as within-subjects factors compared the proportion of common source reports in 26 control and 25 ASD subjects.

Further, to quantify putative differences in how ASD and control subjects inferred the causal relationship between auditory and visual stimuli, Gaussian functions were fit to the proportion of common source reports as a function of Δ (e.g. *Rohe and Noppeney, 2015*). These fits yielded three parameters of interest: (1) amplitude (tendency to report common cause when maximal), (2) mean (spatial disparity at which auditory and visual cues are most likely considered to originate from a common cause), and (3) width (spatial disparity range over which subjects are likely to report common cause).

## Experiment 4: Audio-visual common source reports under temporal disparities (Explicit)

Twenty-one ASD (age = 15.94±0.56 years) and 19 control (age = 16.3±0.47 years) subjects participated in this task. As expected, ASD subjects had significantly higher SCQ (ASD: SCQ = 18.31±1;

control: SCQ = 4.92±0.73; t-test: $t_{32}$=–9.41, p<0.0001) and AQ (ASD: AQ = 32.76±1.58; control: AQ = 14.58±1.15; t-test: $t_{32}$=7.43, p<0.0001) scores than the control subjects. Subjects viewed a flash and heard an audio beep (same stimuli as in Experiments 1, 2, and 3) presented centrally either at the same time or at different asynchronies. Twenty-three different temporal disparities (Δ) were presented: 0, ±10, ±20, ±50, ±80, ±100, ±150, ±200, ±250, ±300, ±500, and ±700 ms (positive Δs indicate that flash led the auditory stimulus). Subjects indicated if the flash and beep were synchronous (exact instruction: 'appeared at the same time') or asynchronous ('appeared at different times') via button press on a standard computer mouse. Each Δ was presented 25 times in random order.

Proportion of synchronous reports at each Δ was calculated. A Gaussian function was fit to the proportion of synchronous reports as a function of Δ (ASD: $R^2$=0.86±0.05; control: $R^2$=0.94±0.01). The Gaussian fits yielded three parameters that characterized subjects' performance: (1) amplitude (representing the maximum proportion of synchronous reports), (2) mean (representing the Δ at which subjects maximally perceived the flash and beep to be synchronous), and (3) width (representing the range of Δ within which subjects were likely to perceive the auditory and visual stimuli to co-occur in time).

A mixed-effects ANOVA with group as the between-subjects factor, and temporal disparity (Δ) as a within-subjects factor compared the proportion of synchronous reports. Similarly, independent-samples t-tests compared the parameters of the Gaussian fits between the groups.

## Experiment 5: Visual heading discrimination during concurrent object motion

Fourteen ASD and 17 control subjects (ASD: 15.71±0.5 years; control: 16.3±0.6 years) participated in this task. The ASD group had significantly higher SCQ (ASD: 16.71±1.36; control: SCQ = 7.35±1.12; p<0.0001) and AQ scores (ASD: AQ = 33.78±2.20; control = 11.79±2.35, p<0.0001) than the control group. Details of the apparatus and experimental stimuli have been previously described (**Dokka et al., 2019**).

In brief, subjects viewed lateral movement of a multipart spherical object while presented with a 3D cloud of dots mimicking forward translation (**Figure 2—figure supplement 2A**). The multipart object moved rightward or leftward within a fronto-parallel plane at five peak speeds: 0.07, 0.13, 0.8, 2.67, and 5.33 m/s. Implied self-motion consisted of a single interval, 1 s in duration, during which the motion stimulus followed a smooth Gaussian velocity profile (displacement = 13 cm; peak velocity = 0.26 m/s). Heading was varied in discrete steps around straight forward (0°), using the following set of values: 0, ±5, ±10, ±15, ±20, ±25, and ±45° (positive value indicates rightward heading). In one session, subjects indicated if they perceived the object to be stationary or moving in the world. In another session, subjects indicated if their perceived heading was to the right or left of straight ahead. In each session there were a total of 130 distinct stimulus conditions (2 object motion directions × 5 object motion speeds × 13 headings) and each condition was presented 7 times. All stimulus conditions were interleaved in each block of trials.

Heading discrimination performance was quantified by fitting psychometric curves for each object motion direction and speed (**Dokka et al., 2019**). These fits yielded parameters that characterize the accuracy and precision of heading perception: bias and threshold. For statistical analyses, the bias measured with leftward object motion was multiplied by –1, such that expected biases were all positive (**Dokka et al., 2019**). To quantify the differences in the heading bias between groups, a third-order regression model of the form: y = $b_0$ + $b_1$X + $b_2$X2 + $b_3$X3, where X is the sign consistent logarithm of object motion speed was fitted to the heading bias. We compared the linear ($b_1$), quadratic ($b_2$), and cubic ($b_3$) coefficients along with their corresponding partial $R^2$ values between groups, similar to the analyses performed on the auditory bias in the audio-visual localization tasks.

## Causal Inference Modeling

We modeled subject responses using a causal inference model (**Figure 4A**) where the observer has to infer whether two sensory cues (auditory and visual) come from the same or separate causes(s), and use this information to either integrate or not information from these cues. In each trial, we assume that the subject's observations of the auditory and visual location (denoted $X_a$ and $X_v$) are the experimenter defined veridical values (denoted by $\epsilon_a$ and $\epsilon_v$) corrupted by sensory noise with variances $\sigma_a^2$ and $\sigma_v^2$ ,

$$p\left(X_a|\epsilon_a\right) = \mathfrak{N}\left(X_a; \epsilon_a, \sigma_a^2\right) \tag{3}$$

$$p\left(X_v|\epsilon_v\right) = \mathfrak{N}\left(X_v; \epsilon_v, \sigma_v^2\right) \tag{4}$$

where $\mathfrak{N}\left(x; \mu, \sigma^2\right)$ denotes the normal probability density function with mean μ and variance $\sigma^2$. We assume that subjects have a good estimate of their sensory uncertainties (over lifelong learning) and hence the subject's estimated likelihoods become,

$$l\left(S_a\right) \equiv p\left(X_a|S_a\right) = \mathfrak{N}\left(X_a; S_a, \sigma_a^2\right) \tag{5}$$

$$l\left(S_v\right) \equiv p\left(X_v|S_v\right) = \mathfrak{N}\left(X_v; S_v, \sigma_v^2\right) \tag{6}$$

where $S_a$ and $S_v$ denote the inferred location of auditory and visual stimuli. The subject's joint prior over the cue locations is parameterized as a product of three terms which reflect:

(a) $f_{natural}\left(S_a, S_v\right)$ : the subject's natural prior over the unisensory cue locations. For example, subjects may have a prior that sensory cue locations are more likely to occur closer to midline as compared to peripheral locations. We model this component of the prior as normal distributions where the mean and variance are unknown parameters fitted to the data.

$$f_{natural}\left(S_a, S_v\right) = \mathfrak{N}\left(S_a; \mu_a, \sigma_{ap}^2\right) \mathfrak{N}\left(S_v; \mu_v, \sigma_{vp}^2\right) \tag{7}$$

(b) $f_{CI}\left(S_a, S_v|C\right)$ : the influence that the inferred cause (C) has on the knowledge of cue locations. In our causal inference model $S_a$ is inferred as being equal to $S_v$ if C=1 and independent if C=2.

$$f_{CI}\left(S_a, S_v|C\right) = \begin{cases} \delta(s_a - S_v) & \text{if } C = 1 \\ 1 & \text{if } C = 2 \end{cases} \tag{8}$$

(c) $f_{task}\left(S_a|D\right)$ : the relationship between the inferred trial category (D) and the cue locations.

## Implicit task

In the implicit discrimination task, where the trial category corresponds to the side of the auditory cue location relative to the midline, $S_a$ is positive if $D_{imp} = 1$ and negative if $D_{imp} = -1$.

$$f_{task}\left(S_a, S_v|D_{imp}\right) = \begin{cases} H\left(S_a\right) & \text{if } D_{imp}=1 \\ H\left(-S_a\right) & \text{if } D_{imp}=-1 \end{cases} \tag{9}$$

where H(x) is the Heaviside function (H(x)=1 if x>0 and 0 otherwise).

The product of *Equations 7–9*, defines the probability over cue locations conditioned on C and $D_{imp}$ in the implicit task as

$$p_{implicit}\left(S_a, S_v|C, D_{imp}\right) \propto f_{natural}\left(S_a, S_v\right) f_{CI}\left(S_a, S_v|C\right) f_{task}\left(S_a, S_v|D_{imp}\right) \tag{10}$$

which can be succinctly written as

$$p_{implicit}\left(S_a, S_v|D_{imp}, C\right) \propto \mathfrak{N}\left(S_a; \mu_a, \sigma_{ap}^2\right) \mathfrak{N}\left(S_v; \mu_v, \sigma_{vp}^2\right) \left[(C-1) + (2-C)\,\delta\left(S_a - S_v\right)\right] H\left(D_{imp}S_a\right) \tag{11}$$

We parameterize the observer's priors over $D_{imp}$ and C as Bernoulli distributions with means pchoice and $p_{common}$.

$$p_{implicit}\left(D_{imp} = 1\right) = Ber\left(D_{imp}; p_{choice}^{implicit}\right) \tag{12}$$

$$p\left(C = 1\right) = Ber\left(C; \text{pcommon}\right) \tag{13}$$

The posterior probability of the subject inferring the auditory cue to come from the right can be obtained by marginalizing over the observer's belief whether the auditory and visual cue come from a single or from separate causes

$$p_{\text{implicit}}\left(D_{\text{imp}} = 1|X_a, X_v\right) = \sum_{c\in\{1,2\}} p_{\text{implicit}}\left(D_{\text{imp}} = 1|X_a, X_v, C = c\right) p\left(C = c|X_a, X_v\right) \tag{14}$$

We assume the subject makes their response by choosing the response that has the highest posterior probability. If $R_{\text{implicit}}$ is the subject response (1 for right and –1 for left), then

$$R_{\text{implicit}} = \arg\max_{d\in\{-1,1\}} p_{\text{implicit}}\left(D_{\text{imp}} = d|X_a, X_v\right) \tag{15}$$

## Explicit task

We model the explicit task by assuming that the decision maker computes the belief over the trial category $D_{\text{exp}}$ using the inferred belief over C, but not exactly equating both (graphical model in *Figure 4A*). This extends earlier approaches (*Körding et al., 2007*) which equate trial category $D_{\text{exp}}$ with C, and additionally allows us to model task specific beliefs about the trial category. As we will show later, such a difference in beliefs between $D_{\text{exp}}$ and C is mathematically equivalent to the subject making their decision by comparing their belief over C to a criterion different from 0.5.

The subject's knowledge about the relationship between the trial category and the inferred variable C is parameterized as $\alpha_{\text{task}}$, as given by *Equation 16* and *Equation 17*

$$p\left(C = 1|D = 1\right) = Ber\left[C; \text{pcommon} + \alpha_{\text{task}}\left(1 - \text{pcommon}\right)\right] \tag{16}$$

$$p\left(C = 1|D = 2\right) = Ber\left[C; \text{pcommon} - \alpha_{\text{task}}\left(\text{pcommon}\right)\right] \tag{17}$$

For $\alpha_{\text{task}} = 0$ there is no relationship between trial category D and C (e.g. before learning the task), and thus the prior over C reduces to pcommon. On the other extreme, $\alpha_{\text{task}} = 1$ corresponds to complete task-learning, where C and $D_{\text{exp}}$ are identical.

The prior probability of the subject's belief over $D_{\text{exp}}$ in the explicit task is parameterized as a Bernoulli distribution with mean pchoice as given in *Equation 18*

$$p_{\text{explicit}}\left(D = 1\right) = Ber\left(D; p_{\text{choice}}^{\text{explicit}}\right) \tag{18}$$

We modeled subject's belief about the sensory cue locations as the product of two terms: $f_{natural}\left(S_a, S_v\right)$ and $f_{CI}\left(S_a, S_v|C\right)$ (*Equation 7* and *Equation 8*)

$$p_{explicit}\left(S_a, S_v|C\right) \propto f_{natural}\left(S_a, S_v\right) f_{CI}\left(S_a, S_v|C\right)$$

$$p_{explicit}\left(S_a, S_v|C\right) \propto \begin{cases} f_{natural}\left(S_a, S_v\right) \delta\left(S_a - S_v\right), & \text{if } C=1 \\ f_{natural}\left(S_a, S_v\right), & \text{if } C=2 \end{cases} \tag{19}$$

with appropriate normalization constants obtained by integrating over all $S_a$ and $S_v$, we get

$$p_{explicit}\left(S_a, S_v|C\right) = \begin{cases} \dfrac{\mathfrak{N}\left(S_a; \mu_a, \sigma_{ap}^2\right)\mathfrak{N}\left(S_v; \mu_v, \sigma_{vp}^2\right)}{\mathfrak{N}\left(\mu_a; \mu_v, \sigma_{ap}^2+\sigma_{vp}^2\right)}\delta\left(S_a - S_v\right) & \text{if } C=1 \\ N\left(S_a; \mu_a, \sigma_{ap}^2\right) N\left(S_v; \mu_v, \sigma_{vp}^2\right) & \text{if } C=2 \end{cases} \tag{20}$$

Our model makes choice $R_{\text{explicit}} = 1$ if

$$p_{explicit}\left(D = 1|X_a, X_v\right) > p_{explicit}\left(D = 2|X_a, X_v\right) \tag{21}$$

which by Bayes rule reduces to,

$$p_{explicit}\left(X_a, X_v|D = 1\right) p_{\text{choice}}^{\text{explicit}} > p_{explicit}\left(X_a, X_v|D = 2\right)\left(1 - p_{\text{choice}}^{\text{explicit}}\right) \tag{22}$$

where the likelihood over observations is evaluated by marginalizing across inferred sensory locations using the sensory likelihoods (*Equation 5* and *Equation 6*), i.e.,

$$p_{explicit}\left(X_a, X_v | C = c\right) = \int \int p\left(X_a, X_v | S_a, S_v\right) p_{explicit}\left(S_a, S_v | C = c\right) dS_a dS_v \tag{23}$$

We can marginalize out C in *Equation 22* to get

$$
\begin{aligned}
&p_{\text{choice}}^{\text{explicit}}\, p_{explicit}\left(X_a, X_v | C = 1\right)\left[\text{pcommon} + \alpha_{\text{task}}\left(1 - \text{pcommon}\right)\right] + p_{\text{choice}}^{\text{explicit}} \\
&p_{explicit}\left(X_a, X_v | C = 2\right)\left[1 - \text{pcommon} - \alpha_{\text{task}}\left(1 - \text{pcommon}\right)\right] > \\
&\left(1 - p_{\text{choice}}^{\text{explicit}}\right) p_{explicit}\left(X_a, X_v | C = 1\right)\left[\text{pcommon} - \alpha_{\text{task}}\left(\text{pcommon}\right)\right] \\
&+ \left(1 - p_{\text{choice}}^{\text{explicit}}\right) p_{explicit}\left(X_a, X_v | C = 2\right)\left[1 - \text{pcommon} + \alpha_{\text{task}}\left(\text{pcommon}\right)\right]
\end{aligned}
\tag{24}
$$

By combining terms, *Equation 24* can be simplified as

$$p_{explicit}\left(\text{X}_a, \text{X}_v | C = 1\right) p_{\text{combined}} > p_{explicit}\left(\text{X}_a, \text{X}_v | C = 2\right)\left(1 - p_{\text{combined}}\right) \tag{25}$$

where $p_{\text{combined}}$ is a function of pcommon , $p_{\text{choice}}^{\text{explicit}}$ and $\alpha_{\text{task}}$ as given in *Equation 26* which cannot be individually constrained.

$$p_{\text{combined}} = \max(0, min(1, \frac{p_{\text{common}}(2p_{choice}^{explicit}-1)+\alpha_{task}[p_{\text{common}}(1-p_{choice}^{explicit}+p_{choice}^{explicit}(1-p_{\text{common}}))]}{(2p_{common}-1)(2p_{choice}^{explicit}-1)+2\alpha_{task}[p_{common}(1-p_{choice}^{explicit})+p_{choice}^{explicit}(1-p_{common})]})) \tag{26}$$

We now show that a decision rule as given in *Equation 26* is equivalent to a subject making their decision by comparing their inferred posterior $p_{explicit}\left(C = 1 | X_a, X_v\right)$ to a criterion t, i.e., $R_{explicit}$ =1 if

$$p_{explicit}\left(C = 1 | X_a, X_v\right) > t \tag{27}$$

Or equivalently

$$p_{explicit}\left(C = 1 | X_a, X_v\right)\left(1 - t\right) > p_{explicit}\left(C = 2 | X_a, X_v\right) t \tag{28}$$

which can be expanded using Bayes rule as given in *Equation 29*

$$p_{explicit}\left(X_a, X_v | C = 1\right)\left(1 - t\right) \text{pcommon} > p_{explicit}\left(X_a, X_v | C = 2\right)\left(t\right)\left(1 - \text{pcommon}\right) \tag{29}$$

Comparing *Equation 29* to *Equation 25*, we can relate terms to get

$$p_{\text{combined}} = \frac{(1-t)\,\text{pcommon}}{(1-t)\,\text{pcommon}+(t)\,(1-\text{pcommon})} \tag{30}$$

where the criterion t is a function of pcommon, $p_{\text{choice}}^{\text{explicit}}$ and $\alpha_{\text{task}}$ .

We provide further model derivation and fitting details in *Supplementary Materials*, *Supplementary file 3*, *Supplementary file 4*. We can also similarly derive the causal inference model for the simultaneity judgement by modeling the temporal percepts as Bayesian inference and replacing the spatial disparities with temporal disparities. Further details are provided in the *Supplementary Materials*, (*Supplementary file 5*).

Last, as a contrast to the causal inference model (and variants thereof, alternatives A–D presented in the main text), for explicit tasks we also fit a functional form, specified by a Gaussian (mean and SD as free parameters) plus an additive bias (*Figure 3—figure supplement 3*). We fit this model to the spatial common cause reports (*Figure 3A*) of control subject. Then, we vary the additive bias, *b* (see *Figure 3—figure supplement 3*), in attempting to account for the ASD data relative to the control. Both the fit to the control data, and to the ASD data relative to the control, were better accounted for by the causal inference model (which additionally is a principled one), than the functional form.

## Acknowledgements

We thank Jing Lin and Jian Chen for programming the experimental stimulus. This work was supported by NIH U19NS118246 (to RH and DEA), and by the Simons Foundation, SFARI Grant 396,921 and Grant 542949-SCGB (to DEA).

# Additional information

## Funding

| Funder | Grant reference number | Author |
|---|---|---|
| National Institutes of Health | NIH U19NS118246 | Dora E Angelaki<br>Ralf M Haefner |
| Simons Foundation Autism Research Initiative | 396921 | Dora E Angelaki |

The funders had no role in study design, data collection and interpretation, or the decision to submit the work for publication.

## Author contributions

Jean-Paul Noel, Conceptualization, Data curation, Formal analysis, Investigation, Methodology, Visualization, Writing - original draft, Writing – review and editing; Sabyasachi Shivkumar, Conceptualization, Data curation, Formal analysis, Investigation, Methodology, Software, Visualization, Writing – review and editing; Kalpana Dokka, Conceptualization, Data curation, Investigation, Methodology; Ralf M Haefner, Conceptualization, Funding acquisition, Investigation, Methodology, Project administration, Resources, Supervision, Writing – review and editing; Dora E Angelaki, Conceptualization, Funding acquisition, Project administration, Supervision, Writing – review and editing

## Author ORCIDs

Jean-Paul Noel http://orcid.org/0000-0001-5297-3363
Ralf M Haefner http://orcid.org/0000-0002-5031-0379
Dora E Angelaki http://orcid.org/0000-0002-9650-8962

## Ethics

Human subjects: The study was approved by the Institutional Review Board at the Baylor College of Medicine (protocol number H-29411) and written consent/assent was obtained.

## Decision letter and Author response

Decision letter https://doi.org/10.7554/eLife.71866.sa1
Author response https://doi.org/10.7554/eLife.71866.sa2

# Additional files

**Supplementary files**

• Supplementary file 1. Control participants. The table indexes each control participant by a unit ID, and indicates in which experiment did each participant take part in. Green indicates that the participant took part in the given experiment, while red indicates that they did not. Experiment 1 is the unisensory discrimination task and audio-visual cue combination that does not require causal inference (i.e. imperceptible disparities). Experiment 2 is the audio-visual implicit causal inference experiment. Experiment 3 is the explicit causal inference experiment with spatial disparities. Experiment 4 is the explicit causal inference experiment with temporal disparities. Experiment 5 is the visual heading discrimination task during concurrent object-motion.

• Supplementary file 2. ASD participants. The table indexes each ASD participant by a unit ID, and indicates in which experiment did each participant take part in. Green indicates that the participant took part in the given experiment, while red indicates that they did not. An orange box indicates that the participant took part in the experiment, but their data was excluded in presentation of the empirical results (but not the modeling of individual subjects, as indicated in the main text). Experiment 1 is the unisensory discrimination task and audio-visual cue combination that does not require causal inference (i.e. imperceptible disparities). Experiment 2 is the audio-visual implicit causal inference experiment. Experiment 3 is the explicit causal inference experiment with spatial disparities. Experiment 4 is the explicit causal inference experiment with temporal disparities. Experiment 5 is the visual heading discrimination task during concurrent object-motion.

• Supplementary file 3. Priors distributions over model parameters.

• Supplementary file 4. Causal inference model inference and fitting details.

- Supplementary file 5. Causal inference modeling for simultaneity judgments.
- Transparent reporting form

## Data availability

Data and code are available at https://osf.io/6xbzt.

The following dataset was generated:

| Author(s) | Year | Dataset title | Dataset URL | Database and Identifier |
|---|---|---|---|---|
| Noel S, Dokka HA | 2022 | ASD Causal Inference | https://osf.io/6xbzt | Open Science Framework, 6xbzt |

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
