## [Editor Report]

Autism spectrum disorder is characterized by social, communicative and sensory anomalies. This study uses behavioral psychophysics experiments and computational modelling to interrogate how individuals with autism combine sensory cues in multisensory tasks. The results showed that individuals with autism were more likely to integrate cues, but less likely to report doing so, thus raising interesting questions regarding how individuals with autism perceive the world.

---

## [Decision Letter]

**Decision letter after peer review:**

Thank you for submitting your article "Aberrant causal inference and presence of a compensatory mechanism in Autism Spectrum Disorder" for consideration by *eLife*. Your article has been reviewed by 3 peer reviewers, and the evaluation has been overseen by a Reviewing Editor and Richard Ivry as the Senior Editor. The following individual involved in review of your submission has agreed to reveal their identity: Ulrik Beierholm (Reviewer #3).

Essential revisions:

1) The experiments report interesting results regarding audio-visual integration for spatial discriminations in both typical individuals and individuals with ASD. However, the conceptual framing (including the model) is one of several potential accounts of these data, and should be framed as such. Alternative accounts need to be presented and seriously discussed, and not just as an extension of the Discussion. The abstract and other parts of the manuscript also need to be adjusted accordingly.

2) Related to point 1. Prominent aspects of the data, including higher overall bias in autism in Figure 2, are not captured in the model in Figure 4. The dissociation between explicit and implicit is not convincing, and the stress on group differences puts an emphasis on small effects. Please revise model and/or Discussion to address these concerns.

3) Model fitting is not described sufficiently. How were the sensory parameters fitted? It seems that more than 20 parameters were fitted (Supp. File 1) for the aggregate subject through the slice sampling, is that correct? Was this also done for individual subjects? What was done to ensure convergence? Was any model comparison done? Please include a list or figure showing the different steps of the model fitting.

4) The model may be over-specified with both a lapse rate and a lapse bias. Please test a simpler model without lapse bias or explain why that was not done.

5) In experiments 3 and 4 please detail the specific instructions given. Specifically, were participants asked to press a button if they thought both cues come from the same source, or if they thought that the 2 cues come from 2 sources? Since there was not a default option (an "I don't know option"), it's important to know the default – determined by the way the question was phrased.

6) The participants in each experiment were not clearly described. Please provide more details about the task completion of participants, such as how many completed all four tasks, etc. A table would be helpful. Specifically, what were the performance scores in Experiment 1 – of the sub-group of participants of Experiment 2 – the question of whether the psychometric plots did not differ between ASD and controls participating in this study is crucial for estimating whether they were expected to have different magnitudes of bias (as they actually did). The authors did not address the question of the overall bias magnitude, only the values at the large disparities.

7) Please specify the criteria for the ASD diagnosis, DSM-5 or DSM-4? Are they classic autism or Asperger or PDD-NOS subjects? Were the gold standard ADOS ADIR performed to confirm the diagnosis? If not, the authors should acknowledge this as a limitation in Discussion.

8) More detailed research participant description is required. SCQ and AQ were performed for all participants. Were there ASD individuals below the cut-off of these two scales? or any TD participants above the cut-off? This information should be stated. The authors should consider excluding the ASD individuals below the cut-offs and TD individuals above the cut-offs from data analysis. Please provide more details about how the TD participants were recruited. IQ was available for a subset of the ASD participants: How many of them have IQ scores? IQ was measured using what test? Was the IQ measured for the TD group?

9) Please report effect sizes, e.g. eta^2^ or Cohen's d.

*Reviewer #1:*

Using a series of cue combination tasks, the authors studied the causal inference of multisensory stimuli in people with ASD. The authors found the intact ability in optimal cue combination of participants with ASD but impairment in dissociating audio and visual stimuli when presented with wider spatial disparity. It suggested they persisted with a wrong integration model for causal inference. However, the individuals with ASD explicitly report the common cause of stimuli fewer than the controls. Through formal modeling, the authors found increased prior probability for the common cause in ASD. However, reporting the common cause in ASD is reduced in the explicit task, indicative of a compensatory mechanism via a choice bias.

In general, I think this study was well-designed and the results were interesting. The conclusions of this paper are mostly well supported by data. But I have a few questions that I would like to see the author’s address.

1. When comparing the temporal disparity task to the spatial task, the authors concluded that the overall reduced tendency to report common cause at any disparity and across spatial and temporal conflicts seemingly is the defining characteristic of ASD. However, in Figure 3D, it could tell that a higher proportion of common cause reporting in ASD when absolute temporal disparity became greater, which differed from the case of spatial task and from when the temporal disparity was narrower. Could the conclusion be too general? The authors should tone it down or give more discussion about the incongruence.

2. When fitting the model to individual subject data, the authors found comparable p_combined_ for the explicit task between ASD and control subjects. This seemed to be contrasted to the result of aggregate data and behavioral results. Did the difference come from the fitting procedure? Did the significant decreased in p_combined_ was because of the lack of consideration of subject heterogeneity? The authors could provide more explanation or discussion of it.

3. A related question is about the intuition behind the two steps of modeling fitting (i.e., to aggregate and individual data). What more could fitting models to aggregate or individual data provide to one another procedure? The authors should elaborate on it.

4. I would like to see the authors discuss more the interesting finding of a potential compensatory mechanism, particularly the meaning of it in terms of the possible relation to ASD symptoms. For example, how would the increased prior probability of common cause report and the compensatory choice bias contribute to the sensory abnormalities in ASD?

5. The participants in each experiment were not clearly introduced. The authors should provide more details about the task completion of participants, such as how many completed all four tasks, etc. And the data of how many participants who participated in both the implicit and explicit spatial task were included in modeling?

6. The authors could also conduct some correlational analyses between estimated model parameters and symptomatology measures, just as what they have done for psychometric features, to further investigate how autistic symptoms would affect the process of causal inference.

7. Since the data of the individuals with poor performance were also fitted (such as 8 of the individuals with ASD in Experiment 3), it is interesting to see if there is anything special or atypical in terms of their model parameters, even though their data were not included in behavioral analyses.

8. I suggest specifying the criteria for the ASD diagnosis, DSM-5? or DSM-4? or ICD-10? Are they classic autism or Asperger or PDD-NOS? Were the gold standard ADOS ADIR performed to confirm the diagnosis? If not, the authors should acknowledge this as the limitation in Discussion.

9. SCQ and AQ were performed to all participants. My question is: is there any ASD individuals below the cut-off of these two scales? or any TD participants above the cut-off. the authors should consider excluding the ASD individuals below the cut-offs and TD individuals above the cut-offs from the data analysis.

10. Please provide more details about how the TD participants were recruited?

11. IQ was available for a subset of the ASD participants: How many of them have IQ scores? Is there any particular reason that the other ASD participants did not have IQ scores? How the IQ was measured? using Wechesler or Raven's test? Was the IQ measured for the TD group?

12. The authors could provide direct comparisons of thresholds and visual weights between two groups in the result section of Experiment 1.

13. Errors bars in Figure 1E and 1H were not very obvious. The authors could consider using simpler markers, such as "+" (i.e., short lines) for simultaneously displaying horizontal and vertical error bars.

14. It should be "As for the case of auditory disparities, …" instead of " As for the case of spatial disparities, …" for the first sentence of the second paragraph after Figure 3.

*Reviewer #2:*

The paper consists of 4 interesting experiments examining multisensory processing in autism spectrum disorder. The first experiment shows that participants with ASD perform similar to controls in cross-model integration, a conceptual replication of earlier findings from this group. However, the subsequent experiments reveal some intriguing differences between the groups in terms of how they use explicit and implicit information in evaluating if auditory and visual information comes from a common source or distinct sources. The authors propose a model that aims to explain the seeming dissociation between explicit and implicit reports of the two groups. The strength of this work is that the experiments are very interesting and report interesting results regarding audio-visual integration for spatial discriminations in both typical individuals and people with ASD. The comparison between explicit and implicit reports is very interesting. In terms of weaknesses, the dissociation between explicit and implicit is not convincing, and the stress on group differences puts an emphasis on, at best, marginal effects, which the modelling does not explain. For example, an alternative account that is consistent with all the data presented is that there are individuals with ASD who are somewhat poorer auditory discriminators, resulting in the bias effects and broader disparities. These individuals would be less likely to commit to an explicit "single source" statement in line with their reduced auditory localization skills.

The dissociation between explicit and implicit is not convincing, and the stress on group differences puts an emphasis on, at best, marginal effects, which the modelling does not explain (the strongest linearity on ASD's curve in Figure 2 – is not captured in the modelling in Figure 4) For example, an alternative account that is consistent with all the data presented is that there are individuals with ASD who are somewhat poorer auditory discriminators and they impacted overall performance in Experiment 2, resulting in a larger bias effect, and also somewhat broader in disparities. These individuals would be less likely to commit to an explicit "single source" statement, which is quite committing, in line with their reduced auditory localization skills. The authors should at least address this alternative account, and present auditory discrimination curves of Experiment 2's participants.

The model does not account for the data point of individuals with autism being pulled by a reliable visual blob 24 degrees away, which was the main point in Figure 3.

Overall the authors ignore more prominent aspects of the data (e.g. higher overall bias in autism in Figure 2) for points they want to make (non linearity larger in autism than in controls).

Reliability – is a confusing term. The stimuli are reliably presented, but the information the perceivers derive regarding their position is less reliable when stimuli are small.

Figure 1f, g – I had difficulties understanding. I assume that the dashed lines should be to the right of the solid lines, which is the case for "high-reliability" blob, but why is it switched for the low reliability case? In both sample participants (f and g) and I wonder why the bias is larger (larger distance between dashed and matched solid plot, in both participants) for low versus intermediate size (reliability) blobs. If this is the actual result – it needs explanation.

Figure 2 – the main observation is that the bias in autism is larger. Perhaps this group difference stems from this group being somewhat poorer auditory spatial discriminators than their 15 age-matched controls in the experiment. If their auditory discrimination is poorer we would expect an overall larger bias, and perhaps also across a broader range of audio-visual disparities.

Importantly, this is probable account, since this is a smaller population than in Experiment 1 – and their discrimination thresholds are not addressed. Importantly – I could not figure out the overlap in participation across the various experiments. In experiment 1 matched performance was only obtained when 6 participants with ASD were excluded. In Experiment 3 (24 participants originally) – they also excluded a large subgroup, whose behavior was different. Here the group is initially small so variability across participants was not discussed.

The strongest point for the claim of too broad integration is the bottom left point – where high reliability blob has an effect that even increases when the visual blob is presented 24 degrees apart. This point is hard to reconcile (and is not reconciled by the model proposed in Figure 4 either). The authors should show that it is a reliable data point – perhaps by showing single subject data.

In experiments 3 and 4 the specific instructions are crucial – are participants asked to press a specific button if they are perceived as coming from the same source? Or press a button if they are perceived as coming from 2 separate sources. Here phrasing may have affected the decisions of individuals with autism. In order to dissociate between these 2 options it would have been nice to have a third option "don't know". If participants with autism tend to say to be less decisive they would tend to commit to a single source. This account may be explained by being somewhat implicitly poorer localizers.

If you have discrimination functions of the specific subgroups that took part in Experiments 2-3 (since they all participated in Experiment 1 – right?) – please show them or report discrimination skills for these subgroups, since this is the relevant control-ASD matching.

Re modelling and Figure 4 – It is difficult to follow the model – perhaps label the model parameters in the diagram of Figure 4a.

*Reviewer #3:*

In this paper Noel et al., use a combination of psychophysical experiment and computational modeling to examine the differences in behaviour between participant on the Autism Spectrum Disorder and control participants when dealing with multi-sensory stimuli (e.g. audio-visual). It is well known that ASD subjects tend to differ in how they combine such stimuli, and it has previously been suggested that this may be due to a difference in the tendency to perform causal inference.

The study indeed finds that while ASD participants had similar ability to combine cues when unambiguously from the same source, they differed in the tendency to combine them when unclear if necessary to combine. In contrast when asked to explicitly indicate whether stimuli originated from the same source (and therefore should be combined) they tended to under report.

While the experiments are in themselves very standard, the paper relies on computational modeling to differentiate the possible behavioural effects, using advanced Bayesian statistical methods.

These results confirm existing ideas, and build on our understanding of ASD, while still leaving many questions unanswered. The results should be of interest to anyone studying ASD as well as any other developmental disorders, and perception in general.

I enjoyed reading this paper, although the model fitting procedure especially was not clear to me. How were the sensory parameters fitted? By my count more than 20 parameters were fitted (Supp. File 1) for the aggregate subject through the slice sampling, is that correct? Was this also done for individual subjects? I would be nervous about fitting that many parameters for individual subject data. What was done to ensure convergence?

Was any model comparison done? Might be better to include a list or figure showing the different steps of the model fitting.

I also worry that the model is over specified with both a lapse rate and a lapse bias. From my understanding the lapse rate specifies when subjects (through lack of concentration or otherwise) fail to take trial stimuli into account and therefore go with their prior. In other studies this prior may be identical to the prior over spatial range, or may be a uniform discrete distribution over the bottoms available for response.

Maybe the variables are constrained in ways that I did not understand, but with just a binary response (Left/Right) the model can largely incorporate any bias to a large set of possible parameter values of lapse rate and bias. I.e. that the model is over specified. That would also explain the wide range of values for the fitted parameters in Figure 3.

I think this should really be investigated before the results can be trusted.

Looking at Figure 4E and F makes me hesitant about trusting the results.

Authors also acknowledge that the lapse bias and P combined are too closely entwined to really be well separated in the explicit temporal experiment. Maybe for that reason it would also be useful to test a simpler model without lapse bias?

I find it mildly confusing that D refers to a Left/Right response in the implicit task, and Common/Separate in the explicit task. Maybe better to use separate symbols? D is fine for 'decision' but in places in the text it is instead referred to as 'trial category' which is vague. I also don't really think D is needed in the generative model in Figure 4 as it is not really causing the subsequent variables C or Sa.

Does *eLife* not require the reporting of effect sizes (e.g. eta^2^ or Cohen's d)? It would be good to include these.

The plots in Figure 3 mostly look like shifts up for ASD relative to controls. The authors might want to fit a model with a positive bias, i.e.

a*N(mu,sd^2^)+b

may fit better (could do model comparison) and just show difference in b. This is just a suggestion though, but it may be cleaner for their argument.

In the Discussion, while divisive normalisation is one way to achieve the marginalisation needed for Bayesian causal inference, there are other ways to achieve it (Cuppino et al., 2017, Yamashita 2013, Yu et al., 2016, Zhang et al., 2019). It would be good to acknowledge this.

Eq 5 and 6, 38 are misleading. Likelihood is a function of Sa/Sv, so would be better to write as l(Sa)=N(Xa;Sa,Sv)

Eq 9: is D either 1 or 2? Or 1 or -1?

Detail: maybe use different symbols for lapse rate and lapse bias? I find λ and odell confusing. How about P_lapse_ for the lapse rate to emphasise that it is a probability? P_common_ is already a fitted variable that is also a probability of a Bernoulli distribution.

Page 5 (pages of the pdf):

“ …ASD did not show impairments in integrating perceptually congruent auditory and visual stimuli.”

– “ …ASD did not show impairments in integrating perceptually congruent (and near-congruent) auditory and visual stimuli.”

In experiment 2 there was a six degree discrepancy, so near-congruent seems appropriate.

Typos:

“We perform the integral in Eq. S5 for the implicit task by”: should this be Eq. 35?

References:

Cuppini, C., Shams, L., Magosso, E. and Ursino, M. A biologically inspired neurocomputational model for audiovisual integration and causal inference. Eur. J. Neurosci. 46, 2481-2498 (2017).

Yamashita, I., Katahira, K., Igarashi, Y., Okanoya, K. and Okada, M. Recurrent network for multisensory integration-identification of common sources of audiovisual stimuli. Front. Comput. Neurosci. 7, (2013).

Yu, Z., Chen, F., Dong, J. and Dai, Q. Sampling-based causal inference in cue combination and its neural implementation. Neurocomputing 175, 155-165 (2016).

Zhang, W., Wu, S., Doiron, B. and Lee, T. S. A Normative Theory for Causal Inference and Bayes Factor Computation in Neural Circuits. Adv. Neural Inf. Process. Syst. 32, 3804-3813 (2019).

[Editors’ note: further revisions were suggested prior to acceptance, as described below.]

Thank you for resubmitting your work entitled “Aberrant causal inference and presence of a compensatory mechanism in Autism Spectrum Disorder” for further consideration by *eLife*. Your revised article has been evaluated by Barbara Shinn-Cunningham (Senior Editor) and a Reviewing Editor.

All reviewers agree that the manuscript has improved significantly during revision, but there are some remaining issues to be addressed, as noted below and described in detail in the individual reviews:

1. More detailed description of how statistical analysis was carried out, including clarifications/modifications as suggested by reviewer 1.

2. Rebalancing interpretation of the experimental and odelling results, as suggested by reviewer 2.

*Reviewer #1:*

The authors have addressed my recommendations and questions in much detail. Their changes have improved the quality of the manuscript as a result, illuminating the perceptual causal inference in ASD across different contexts. However, I believe there still are a couple of points that the authors can address to make the description of the results and the methods even clearer for publication.

1. Figure legends/captions of Figures 3 and 4 in the main texts lack detailed descriptions of the elements in the figures. For example, for Figures 3 and 4, what do those error bars represent? Standard errors or confidence intervals? In Figure 4B, are solid lines the model predictions and hollow points the observations? I believe this essential information would help readers better understand the figures.

2. The data points in Figure 2A-B and Figure 3A-C are slightly different from those in Figure 4B-C. For example, in Figure 2B, the audio bias of 24 deg disparity is weaker than that of 12 deg disparity for the high visual reliability condition (dark brown lines and points); however, in Figure 4B left panel, the audio bias of 24 deg disparity is even larger than that of 12 deg disparity. I assume that the data points depicted in Figure 4B-C are the aggregate data for modelling, in which the data of some participants were not included? I notice that the authors have included which participants were included in the single-subject modelling, but was the aggregate data the same as what was used for plotting Figures 2 and 3? I find it a bit confusing at first sight, perhaps the author could check it again and/or mention the related information in the caption or the main text?

3. From lines 451-453 of merged files (Instead, differences between […] relative to control observers.), did the author imply that the model where p_common_ was freely estimated from the data was better, compared with the model where p_common_ was fixed (I guess it’s the model in Figure 4 – supplement 2)? In other words, did the authors have two different models and conduct a model comparison here? If so, I think it’s better to provide model comparison results. The question also applies to the texts from lines 460-461. Also, what is DAIC? Is it the difference of AIC between the full model (that allows p_common_) and the restricted model (that fixes p_common_ to a constant)? The authors should describe it somewhere in the main text.

4. The authors should be more specific about the tests they used to compare model parameters between groups and those correlational analyses. What type of tests did the authors use, parametric (i.e., Welch t-test, Pearson correlation) or non-parametric (i.e., Mann-Whitney, Spearman correlation, or permutation methods)? Particularly for the comparison of p_combined_ (Figure 4G), would the result be different when a non-parametric test was used if the test used in the current revision was parametric? I suggest the authors take more robust approaches given that the distributions of the model parameters seemed not quite Gaussian.

5. What is α and ν in Equation 5 and 6, please define them in the text. Also, it would be better if the authors give a short introduction to the meaning of lapse rate, lapse bias, etc., when mentioning them for the first time. Given that many readers are not very familiar with computational modelling, they may not intuitively understand what these parameters represent.

6. The D in DAIC from line 462 is in another font.

7. I apologize in advance if it’s my mistake but I failed to find Supplementary Text 1 mentioned in lines 430, 451, and 459. Where could I find it?

*Reviewer #2:*

The authors have adequately addressed my comments.

The strong aspects of the results are better clarified, and the overlap between participants across experiments is also clear. Further, the authors do not make claims that are not directly supported experimentally.

The limitation of a somewhat small (<20) number of participants per group in important experiments is still a drawback, given participants’ variability, particularly in the ASD group. Yet, I believe that the main results hold.

The strongest aspects of the study are the direct results, rather than the modelling:

Experiment 1: audio-visual integration is intact in ASD 2. Yet multisensory behavior is atypical (in the current experimental protocol) – ASD participants tend to favor source integration, as manifested by their cross-modal bias in localization even when visual and auditory signal are separable from a sensory perspective. Though both groups tend to over integrate, this is more salient and tend to span a broader distance in ASD. 3. Explicit reports have an opposite tendency – individuals with ASD were less likely to report a common cause for the two stimuli. Given the adequate direct measures of ASD cue integration with a small audio-visual distance (performance in Experiment 1) these results suggest a specific atypicality in cause attribution.

I also find the difference between spatial and temporal integration very interesting. Temporal and spatial groups differences in explicit attribution of a common source merits some additional discussion.

Personally, I think the contribution of the modelling part to the study is overstated in the paper, but I agree that is a personal perspective and need not be imposed on the authors.

*Reviewer #3:*

The authors have done a very good job including new alternative models, and improving the Description of the modelling (modelling my main points of scepticism). I am happy to recommend the paper for publication.

---

## [Author Response]

Essential revisions:1) The experiments report interesting results regarding audio-visual integration for spatial discriminations in both typical individuals and individuals with ASD. However, the conceptual framing (including the model) is one of several potential accounts of these data, and should be framed as such. Alternative accounts need to be presented and seriously discussed, and not just as an extension of the Discussion. The abstract and other parts of the manuscript also need to be adjusted accordingly.

We thank the reviewers for their suggestion and agree that presenting alternative accounts will strengthen the manuscript. We have done this both empirically and via additional modelling.

Empirically, we check whether visual and/or auditory localization performance is equal across the control and ASD cohorts participating in the causal inference task. We now report the auditory and visual discrimination bias and thresholds for these participants (Figure 2 – supplement figure 1). The results show no difference between controls and individuals with ASD, suggesting that a potential baseline difference in sensory processing between these groups does not explain their differing behavior during causal inference.

In additional modelling, we have now included the following potential accounts (see Figure 4 – supplement figure 7, 8, and 9):

A. Forced fusion (all parameters are free to vary, except for C, which is fixed to 1).

B. Forced segregation (all parameters are free to vary, except for C, which is fixed to 2).

C. Uniform lapse bias (Lapse rate, p_common_, p_choice_ are free to vary with a uniform lapse bias – unbiased model).

D. D1. Implicit task: Lapse rate, lapse bias, and p_choice_ are free to vary, others are not.

D2. Explicit task: Since p_choice_ trades off against p_common_ for the explicit task, alternative D2 is similar to D1 (above), but only lapse rate and bias are free to vary.

In alternatives A and B, we fit the model to the ASD aggregate data. In models C and D, we first fit to the control aggregate subject, and then vary the specific parameters noted to fit to the ASD aggregate subject relative to the control. We report AICs for the ASD aggregate subject.

For the implicit task (most cleanly indexing p_common_), all alternative accounts perform worse than the model included in the main text (where lapse rate, lapse bias, p_common_ and p_choice_ are free to vary from control to ASD aggregate subjects; see Figure 4 – supplement figure 7). This suggests that how individuals with ASD infer causal relations, and not their sensory uncertainty, is the most parsimonious factor differentiating between ASD and control (having excluded alternatives A and B that allow for different sensory uncertainties while fixing the causal inference parameter). Further, the fact that alternative A (forced fusion) does better than alternative B (forced segregation) again suggests that individuals with ASD tend to overweight integration relative to segregation. The fact that the model in the main text performs better than Alternative A suggests that individuals with ASD do *not always* integrate, they do perform causal inference, but are biased toward integration over segregation compared to controls.

In response to this comment, we have modified the text to include the unisensory performance for participants in Experiment 2 (P6, “To confirm […] and across all reliabilities*),* as well as the alternative models (P10-11, “Lastly, we consider a set of alternative models […]” vary (main model used in Figure 4)). We have also included new supplementary figures:

– Figure 2 – supplement figure 1. Visual and auditory localization performance of participants in Experiment 2 (audio-visual implicit causal inference).

– Figure 4 – supplement figure 7. Goodness-of-fit of alternative models for the implicit and explicit spatial causal inference task.

– Figure 4 – supplement figure 8. Illustration of the alternative models fits to implicit causal inference model data.

– Figure 4 – supplement figure 9. Illustration of the alternative models fits to explicit causal inference model data.

Lastly, we have included tables specifying which participants took part in the different experiments (in order to examine unisensory performance only of those subjects performing the causal inference task):

– Supplement File 1. Control participants.

– Supplement File 2. ASD participants.

2) Related to point 1. Prominent aspects of the data, including higher overall bias in autism in Figure 2, are not captured in the model in Figure 4. The dissociation between explicit and implicit is not convincing, and the stress on group differences puts an emphasis on small effects. Please revise model and/or Discussion to address these concerns.

The reviewers are correct in pointing out that the model illustrated in Figure 4 (aggregate data) does not fully account for all aspects of the data. However, we must note that we present the aggregate fits to highlight what parameters could or could not in principle explain global differences between the ASD and control cohort. In other words, while the aggregate subjects highlight the common or differing patterns across the two cohorts (ASD and control), even if all subjects used a causal inference strategy, the aggregate subject need not have a pattern completely consistent with the predictions of a causal inference model. On the other hand, the individual subject fits are very good (Figure 4E), with all but one subject showing an explainable variance explained above 80%. To further illustrate the quality of these fits – particularly considering that all experiments are fit jointly! – in this revision we have included supplementary figures showing example control (Figure 4 – supplement figure 3, 4) and ASD (Figure 4 – supplement figure 5, 6) subjects. Again, we must reiterate that we do not fit individual models to account for a particular experiment, but instead attempt to account for a subject’s behavior as a whole, across experiments and sensory modalities.

In the original submission, to highlight global differences between the ASD and control cohorts in a principled manner, we fixed all the sensory parameters (given the results in Experiment 1 showing no unisensory difference across groups) and only allowed for flexibility in the choice and inference parameters. Our new analyses presented in Figure 2 – supplement figure 1, Figure 4 – supplement figure 7 (see above), and Figure 3 – supplement figure 1 (unisensory performance for subjects taking part in Experiment 3)**,** all concord in supporting the fact that there is no difference in sensory performance between the ASD and control cohorts. Nonetheless, we have now also tested this explicitly by testing an alternative model where the sensory uncertainty and choice parameters were allowed to vary from control to ASD (Figure 4 – supplement figure 2) but p_common_ was fixed to the value of the aggregate control (similar to Alternatives A and B above, but where C is fixed to the aggregate control value, as opposed to C=1 or C=2). This model performed worse than that in the main text, where p_common_ and choice parameters were allowed to vary. This indicates that a difference in p_common_ explains better the difference between ASD and control subjects as compared to differences in sensory uncertainty. We also point out that for the individual subjects, sensory uncertainties were fit along with the inference parameters, and we found a significant difference in p_common_ for ASD and control subjects. We found no difference in the estimated sensory uncertainty.

Similarly (and going beyond prior studies) here we fit data from all visual reliabilities jointly, which considerably constrains the model. In Figure 4 – supplement figure 12 (implicit task) and Figure 4 – supplement figure 13 (explicit task) we now demonstrate that either allowing all parameters to vary freely (Panel A), fitting all visual reliabilities separately (Panel B), or doing both of these together (Panel C), would have resulted in better aggregate fits. Even though these alternative models do not concord with our data showing no difference in sensory performance between ASD subjects and controls, these models yield similar results to those reported in the main text, with p_common_ being numerically higher in ASD vs. control subjects, further demonstrating that this (our central) result is highly robust.

Lastly, the reviewers suggest that the group effect is small. To ascertain whether a difference in implicit causal inference between ASD and controls is a robust and replicable effect, and to probe its generalizability beyond audio-visual localization, we have now conducted a conceptual replication and extension. We follow the protocols from Dokka et al., 2019 (*PNAS*), where observers see a pattern of optic flow indicating translation slightly leftward or rightward. Further, they see an independent object, moving at different speeds. Subjects are asked to report their heading (leftward or rightward), and whether the object was stationary or moving. This is a causal inference task, given that if the object is perceived as stationary (hence part of the optic flow), it’s movement on our retinas ought to influence heading perception, but not if the object is perceived to be moving. In Figure 2 – supplement figure 2 (new addition to the revision) we demonstrate that just as for the audio-visual case, control subjects are biased by the independently moving object when this one has a slow speed in the world (similar to the optic flow), but not a fast speed. On the other hand, individuals with ASD seem to readily integrate the independent object into their own heading discrimination, even at large disparities. This is an important conceptual replication of the audio-visual experiment, and thus, even if the effect may be small, it appears to be a reliable one, and a domain general effect.

In response to this comment, we have modified the text to describe the methods (P17, section “Experiment 5: Visual heading discrimination during concurrent object motion”) and results (P6, “Lastly, to further bolster […] segregation (Figure S2C, D).”) of the replication and extension to the implicit causal inference experiment. We have also further described the goodness-of-fit of the aggregate and individual subject fits (P10, “The causal inference model […] as well as implicit and explicit causal inference”).

The following supplementary figures (and appropriate text and figure captions) have been included to (i) illustrate the additional implicit causal inference experiment (Figure 2 – supplement figures 1 and 2), and (ii) explicitly show single subject fits (Figure 4 – supplement figures 3-6) as well as alternative fits (e.g., fitting all visual reliabilities separately, or fitting sensory uncertainty but not p-common, Figure 4 – supplement figures 2, 12, and 13):

– Figure 2 – supplement figure 2. Heading discrimination during concurrent implied self-motion and object motion (Dokka et al., 2019).

– Figure 3 – supplement figure 1. Visual and auditory localization performance of participants in Experiment 3 (audio-visual explicit causal inference).

– Figure 4 – supplement figure 2. Fit to aggregate data for the implicit causal inference task, allowing sensory uncertainty and choice parameters to vary but fixing the inference parameter p_common._

– Figure 4 – supplement figure 3. Data and fit for a single, representative control subject

– Figure 4 – supplement figure 4. Data and fit for another single, representative control subject

– Figure 4 – supplement figure 5. Data and fit for a single, representative ASD subject

– Figure 4 – supplement figure 6. Data and fit for another single, representative ASD subject.

– Figure 4 – supplement figure 12. Fit to aggregate data for the implicit causal inference task, given that all parameters are free to vary (A), the different visual reliabilities are fit separately (B) or both of the above (C).

– Figure 4 – supplement figure 13. Fit to aggregate data for the odellin causal inference task, given that all parameters are free to vary (A), the different visual reliabilities are fit separately (B) or both of the above (C).

3) Model fitting is not described sufficiently. How were the sensory parameters fitted? It seems that more than 20 parameters were fitted (Supp. File 1) for the aggregate subject through the slice sampling, is that correct? Was this also done for individual subjects? What was done to ensure convergence? Was any model comparison done? Please include a list or figure showing the different steps of the model fitting.

We thank the reviewers for their question and apologize this was not clearer in the original submission.

Indeed, each subject had 20 parameters characterizing their responses, yet they were constrained by a large dataset, including four types of tasks (unisensory discrimination, multisensory discrimination, implicit causal inference, and explicit causal inference) and three different visual reliabilities. Further, the prior over the parameters reduces the effective degrees of freedom, such that several parameters have their values tied together (unless constrained to be a specific value based on the empirical data) – equivalent to hierarchical modelling. We have chosen this approach to allow for differences in parameters (e.g., lapse) across experiments that were conducted on different days, but a-priori assuming they are the same. Similarly, when we fit the model to a single (or a subset of) experiment(s), then the parameters specific to the experiment are inferred to be the same as the prior (maximum of prior for the MAP estimate and the prior distribution for the inferred posterior). Therefore, for the aggregate control subject we are effectively fitting 12 parameters for the implicit task and 11 parameters for the explicit task (subset relevant for each task). Moreover, since the model was fit to multiple reliabilities, the prior parameters that were shared across reliabilities were constrained by data from all three reliabilities. For the ASD aggregate subject, either only the choice parameters were varied, or the choice and p_common_ were varied relative to the aggregate control.

We followed a similar procedure for the individual subject fits, but now utilizing their respective empirical data to constrain the parameters. The prior over the parameters is shared as is typical for hierarchical odelling. For the individual subjects, we infer full posteriors over all parameters using slice sampling. The convergence of the sampling was checked by the potential scale factor reduction R^ across chains. Therefore, when we look at the marginal distribution over p_common_ or p_combined_, we marginalize out the other parameters as given by Equation R1. This implicitly performs model averaging by considering different models parameterized by different values of the parameters weighed by their posterior.

p(θinterest|data)=∫p(θinterest|data,θother)p(θother|data)(Equation R1)

In response to the reviewers’ comment, we have summarized the model fitting procedure, both in text (P9-10, “To bridge […] the two experimental groups”) and as a flowchart in the supplementary materials (Figure 4 – supplement figure 1)*.*

4) The model may be over-specified with both a lapse rate and a lapse bias. Please test a simpler model without lapse bias or explain why that was not done.

Allowing for a lapse rate and a lapse bias separately is mathematically equivalent to allowing for two different lapse parameters, each corresponding to a different choice – as is commonly done (e.g., Schütt et al., 2016). We define a prior over lapse bias which is weakly informative, peaking at 0.5. This is the traditional assumption about lapse bias (i.e., unbiased). Therefore, under such a definition, we infer a lapse bias that is different from the commonly assumed 0.5 value if and only if the data has support under such a model. However, as per the reviewers’ suggestion, we have now also tested an alternative (Alternative C in response to Question #1) where only the lapse rate and p_common_ were allowed to vary. As alluded to above, this model had a higher AIC (poorer fit) as compared to also allowing for a lapse bias. For individual subjects, the p_common_ recovered when considering this simpler model was also higher for ASD as compared to control subjects (p=0.008). Similarly, the p_combined_ was not different between the two populations. Thus, the results are consistent with the model allowing for a lapse bias, as presented in the main paper.

5) In experiments 3 and 4 please detail the specific instructions given. Specifically, were participants asked to press a button if they thought both cues come from the same source, or if they thought that the 2 cues come from 2 sources? Since there was not a default option (an "I don't know option"), it's important to know the default – determined by the way the question was phrased.

We thank the reviewers for this question and apologize that we had not provided enough information regarding the instructions. The short answer is that participants were asked to press one button for a given report (e.g., “common cause”) and another for the opposite report (e.g., “separate cause”). Thus, there was no imbalance in the effort required to give one response vs. the other – i.e., there was no default. This important information has now been included in the text (P16, “The exact instructions were […] a standard computer mouse”. And, P17, “Subjects indicated […] standard computer mouse”).

6) The participants in each experiment were not clearly described. Please provide more details about the task completion of participants, such as how many completed all four tasks, etc. A table would be helpful. Specifically, what were the performance scores in Experiment 1 – of the sub-group of participants of Experiment 2 – the question of whether the psychometric plots did not differ between ASD and controls participating in this study is crucial for estimating whether they were expected to have different magnitudes of bias (as they actually did). The authors did not address the question of the overall bias magnitude, only the values at the large disparities.

We have now included in the manuscript Supplement File 1 and Supplement File 2, listing exactly which participants took part in the different experiments (see above). We have also made explicit in the text the inclusion criteria for data in the modeling effort. As suggested by the reviewers, most (but not all, ~80%) participants in Experiments 2 and 3 also took part in Experiment 1, and thus we could detail their visual and auditory localization performance. As we show in Figure 2- supplement figure 1 and Figure 3 – supplement figure 1 (novel additions to the text) there was no difference between the control and ASD sub-groups that participated in Experiments 2 and 3 in terms of their unisensory localization performance. Thus, we can attribute their anomalies in implicit and explicit audio-visual causal inference to the latter computation.

We also wish to highlight the considerable effort it represents to recruit ~40 ASD and ~40 control adolescents to participate in a series of up to 5 experiments (4 in the original manuscript and an additional one added in revisions). Each task took about 60 to 90 minutes to complete and were completed on different days to avoid fatigue. Given that most participants could not transport themselves, on many occasions appointments were missed, rescheduled, and canceled given the caretakers availability. Still, we collected a total of 220 sessions worth of psychophysical data, which represents a significant contribution (particularly when these tasks are informed by a computational approach as is the case here).

7) Please specify the criteria for the ASD diagnosis, DSM-5 or DSM-4? Are they classic autism or Asperger or PDD-NOS subjects? Were the gold standard ADOS ADIR performed to confirm the diagnosis? If not, the authors should acknowledge this as a limitation in Discussion.

The individuals with ASD were diagnosed according to the DSM-5 (which does not distinguish anymore between classic autism and Aspergers). A subset of these subjects (depending on how they were recruited) also counted with an ADOS assessment. There was no statistical difference in AQ, SCQ, or any of the psychometric estimates between ASD individuals with and without ADOS assessment (all p > 0.21). This suggests that all individuals categorized as within the autism spectrum were appropriately diagnosed. We have amended the text to include this information (P14, “Inclusion in the ASD group […] assessment (all p > 0.21).)

8) More detailed research participant description is required. SCQ and AQ were performed for all participants. Were there ASD individuals below the cut-off of these two scales? or any TD participants above the cut-off? This information should be stated. The authors should consider excluding the ASD individuals below the cut-offs and TD individuals above the cut-offs from data analysis. Please provide more details about how the TD participants were recruited. IQ was available for a subset of the ASD participants: How many of them have IQ scores? IQ was measured using what test? Was the IQ measured for the TD group?

There was no individual with ASD below the recommended SCQ cutoff. There were 2 (out of 47, or 4%, i.e., below the false positive rate) control subjects above the recommended cutoff for the SCQ. The AQ is more complicated, given that different cutoffs have been proposed. Most importantly, there were only 3 control subjects with a higher AQ score than the lowest AQ score among all individuals with ASD. All individuals with ASD had AQ scores above the cutoff proposed by 2 out of 3 published recommendations. The text has been modified to include this information (P14, “There was no individual with ASD below […] by Baron-Cohen et al., 2001, cutoff score of 36”).

Excluding the control subjects above the SCQ cutoff / overlapping in AQ score with the ASD subject with the lowest score, did not change the statistical interpretation of any of the reported effects.

The IQ was not measured for the TD group, but we know the mean of the population (by construction) is 100. Thus, we can assume the mean of the TD group was about this value. We only had access to a subset of IQ scores in the ASD cohort (n = 10), because certain clinical providers assessed this metric, while others did not (and thus whether we had an IQ score or not depended on how the subject was recruited). IQ was measured by the Wechsler Adult Intelligence Scale (which is also appropriate for adolescents). The text has been amended to include this information (P14, “Similarly, the Intelligence Quotient […] *mean of 100”*).

9) Please report effect sizes, e.g. eta^2^ or Cohen's d.

Thank you for the suggestion. We now report them throughout the manuscript.

Reviewer #1:Using a series of cue combination tasks, the authors studied the causal inference of multisensory stimuli in people with ASD. The authors found the intact ability in optimal cue combination of participants with ASD but impairment in dissociating audio and visual stimuli when presented with wider spatial disparity. It suggested they persisted with a wrong integration model for causal inference. However, the individuals with ASD explicitly report the common cause of stimuli fewer than the controls. Through formal modeling, the authors found increased prior probability for the common cause in ASD. However, reporting the common cause in ASD is reduced in the explicit task, indicative of a compensatory mechanism via a choice bias.In general, I think this study was well-designed and the results were interesting. The conclusions of this paper are mostly well supported by data. But I have a few questions that I would like to see the author’s address.1. When comparing the temporal disparity task to the spatial task, the authors concluded that the overall reduced tendency to report common cause at any disparity and across spatial and temporal conflicts seemingly is the defining characteristic of ASD. However, in Figure 3D, it could tell that a higher proportion of common cause reporting in ASD when absolute temporal disparity became greater, which differed from the case of spatial task and from when the temporal disparity was narrower. Could the conclusion be too general? The authors should tone it down or give more discussion about the incongruence.

We thank the reviewer for their suggestion. We have now eliminated all reference to the difference in amplitude being the “defining characteristic”. Instead, we weigh equally the fact that individuals with ASD have both a shallower amplitude at small temporal disparities, and a larger width in the Gaussian describing reports of common cause as a function of temporal disparity. The result section has been modified to reflect this change (P9, “As for the case of spatial disparities […] “binding windows”).

In the Discussion section we also eliminated reference to the difference in amplitude as being the “defining characteristic” differentiating ASD from control individuals. We do, however, discuss this finding given that it was present in both the spatial and temporal task (also the visual heading discrimination task included during these revisions) and it is less often acknowledged (relative to the difference in “temporal binding windows”, e.g., Feldman et al., 2018).

Importantly, our modeling strongly implicates the difference in p_common_ as the key difference between ASD subjects and controls in the context of a causal inference task.

2. When fitting the model to individual subject data, the authors found comparable p_combined_ for the explicit task between ASD and control subjects. This seemed to be contrasted to the result of aggregate data and behavioral results. Did the difference come from the fitting procedure? Did the significant decreased in p_combined_ was because of the lack of consideration of subject heterogeneity? The authors could provide more explanation or discussion of it.

The fitting procedure is slightly different, given that in one case we are using aggregate data, while in the other we are fitting to the individual subjects and thus we can leverage knowledge of the subject-specific sensory parameters.

The aggregate subject fitting was performed to highlight what parameters could or could not explain the global differences between ASD and control subjects. To do so, we considered a restricted model class where we assumed matched sensory parameters between ASD and control subjects (based on the findings from Experiment 1, and now Figure 2 – supplement figure 1 and Figure 2 – supplement figure 1). This approach increased the explanatory power of the model by restricting ourselves to a specific model class. This is similar to a bias-variance trade-off whereby introducing a bias (assumption of model family), we can reduce the variance in parameter estimates. Another reason for the aggregate subject having a higher explanatory power is that the aggregate subject has more data (scaled by number of subjects) which also increases the certainty of the estimates. This modelling approach, on the other hand, cannot capture subject heterogeneity, as pointed out by the reviewer. Hence, we also perform the single subject fits.

The individual subject data was fit across experiments and on a per subject level. We allowed for a more flexible model class given that we could estimate also sensory uncertainty parameters. This approach results in better fits (see response to questions above) but increases the heterogeneity in individual parameter estimates.

Overall, we do not think the results are contradictory. First, because slightly different approaches to model fitting were taken. These different approaches maximize the information we may gain, with the aggregate fits attempting to explain global differences between groups and the single subject fits trying to account for individual difference in the observed behavior. Second, there is no conceptual contradiction while there being strong differences in statistical power. In fact, the results are numerically congruent (between aggregate and individual subject). The individual subject data does not differ significantly for P_combined_, but a lack of a statistical difference is not evidence in favor of the null hypothesis. Regardless, even in the individual subject data, the presence of a difference between ASD and control in p_common_, and the lack thereof for P_combined_, indicates a compensatory mechanism since the implicit task clearly demonstrates that ASD individuals are different from controls, but the explicit task, by virtue of being sensitive to compensatory strategies, is not.

We have amended the text to clarify the fitting procedures, and why multiple approaches were taken (P9, “First, we fit aggregate data […] putatively differentiating the two experimental groups”. And, P10, “Overall, both groups were heterogeneous […] Figure 4F and G”). Further, we have added a few sentences in the results addressing the question as to whether aggregate and single subject data fits yielded contrasting results (P10, “Importantly, the aggregate and single subject fits concord in suggesting an explicit compensatory mechanism in individuals with ASD, given that p_common_ is higher in ASD than control (when this parameter can be estimated in isolation) and a measure corrupted by explicit choice biases (i.e., p_combined_) is not.”).

3. A related question is about the intuition behind the two steps of modeling fitting (i.e., to aggregate and individual data). What more could fitting models to aggregate or individual data provide to one another procedure? The authors should elaborate on it.

By inferring the posterior over all model parameters given the data for each individual subject given the data, we are extracting all information there is about each subject, including any individual differences, whether within, or across groups. By repeating this analysis on our aggregate subject, constructed separately for ASD subjects and controls, we now extract information about group-level differences between ASD subjects and controls – the information we are primarily interested in. Additionally, performing the analysis on a group level has the advantage of being able to incorporate knowledge from Experiment 1 that is only available on a group level, namely that ASD subjects and controls have comparable unisensory thresholds – information that by its nature is impossible to use for constraining individual subject fits.

For the aggregate subject analysis, we first combined the data across subjects resulting in a larger dataset. Second, we restricted the model family by assuming that the sensory parameters were the same in the ASD and control population (given empirical observations in Experiment 1) and only the choice and inference parameters were allowed to vary. Of course, in the individual subject data it does not make sense to use estimates from a population (vs. the individual estimates). In turn, the individual subject fits are more flexible and make fewer assumptions, but this results in higher uncertainty in the conclusions, which are drawn from a limited amount of data.

By performing *both* these model fitting procedures, we ensure that we can extract as much information as possible while still ensuring that the conclusions reached are not dependent on assumptions made.

We amended the results (P9, “First, we fit aggregate data […] differentiating the two experimental groups”) to explain why we undertook two modeling approaches and how we interpret the results.

4.I would like to see the authors discuss more the interesting finding of a potential compensatory mechanism, particularly the meaning of it in terms of the possible relation to ASD symptoms. For example, how would the increased prior probability of common cause report and the compensatory choice bias contribute to the sensory abnormalities in ASD?

We thank the reviewer for this suggestion and agree that we could strengthen the clinical impact of this work by discussing a potential relation between the current findings and ASD symptomatology. However, we must also point out that we did attempt correlational analyses between the psychometric effects and two clinical scales (AQ and SCQ). These did not show any reliable correlation, and thus the discussion relating the current findings with symptomatology is speculative (P14, “It is also interesting to speculate […] sensory input ought to be”).

At the same time, a key insight of our analysis is the dissociation between the sensory percept and the behavioral report due to the compensatory bias in ASD subjects. An ASD subject’s sensory percept is determined by p_common_, and therefore differs significantly from controls. The compensatory bias implies an ability (whether conscious or not) by ASD subjects to compensate for this differing percept when responding in the experiment. A key implication of our discovery of this compensatory bias is the fact that the data from explicit tasks cannot be taken at “face value” since they are affected by any compensatory strategy, while the data from implicit tasks doesn’t suffer from this shortcoming.

5. The participants in each experiment were not clearly introduced. The authors should provide more details about the task completion of participants, such as how many completed all four tasks, etc. And the data of how many participants who participated in both the implicit and explicit spatial task were included in modeling?

We have included Supplement File 1 (controls) and Supplement File 2 (ASD) detailing in which experiment or experiments did each subject take part in (see above). Subjects were included in the modeling if they had participated in Experiment 1 (and thus we had an estimate of their sensory encoding) in addition to the particular task of interest. That is, for Figure 4F, we included all participants taking part in Experiments 1 and 2. This included participants deemed poor in Experiment 1, given our attempt to account for participant’s behavior with the causal inference model. For Figure 4G, we included all participants taking part in Experiment 1 and 3.

In addition to Supplement File 1 and Supplement File 2, we have also amended the text (P9, “In a second step […] individual subject behavior.”) to specify the inclusion criteria for the modeling. Similarly, we have amended the caption of Figure 4 to explicitly state which participants were included in the single subject modeling (P13, “Subjects were included […] in Experiment 1 and 3”).

6. The authors could also conduct some correlational analyses between estimated model parameters and symptomatology measures, just as what they have done for psychometric features, to further investigate how autistic symptoms would affect the process of causal inference.

We thank the reviewer for this important suggestion. We attempted correlating the estimated p_common_ and p_combined_ for each subject with their AQ and SCQ measures. None of these correlations (4 in total) was significant. We have amended the text to include this information (P10, “Individual subjects’ p_common_ and p_combined_ as estimated by the model did not correlate with ASD symptomatology, as measured by the AQ and SCQ (all p > 0.17)”).

7. Since the data of the individuals with poor performance were also fitted (such as 8 of the individuals with ASD in Experiment 3), it is interesting to see if there is anything special or atypical in terms of their model parameters, even though their data were not included in behavioral analyses.

We thank the reviewer for this suggestion and agree this is interesting and important information. We looked at the parameters for the ASD subjects who had performed both Experiments 1 and 3, as shown in Figure 4G and Supplement File 2. We z-scored each parameter and looked for outliers, as well as for systematic differences between the ‘good’ (black) and ‘poor’ (red) ASD performers. We classified the parameters as outliers if their absolute z-score exceeded 2. We observed a few outliers (both “good” and “poor” performers) in a parameter or two, but no systematic differences between these sub-groups (see Author response image 1). We have added this information in the text (P10, “Exploration of the model parameters […] causal inference parameters”).

**Author response image 1. sa2fig1:** Z-score of all parameters for all ASD subjects, both included in the main text (black) and not (poor performers, in red). Dashed lines are +/- 2 standard deviations. There are a few outliers, both poor (red) and good (black) performers, but overall there is no categorical difference between sub-groups.

8. I suggest specifying the criteria for the ASD diagnosis, DSM-5? or DSM-4? or ICD-10? Are they classic autism or Asperger or PDD-NOS? Were the gold standard ADOS ADIR performed to confirm the diagnosis? If not, the authors should acknowledge this as the limitation in Discussion.

This has been addressed in Question 7 of the “Essential Revisions.”

9. SCQ and AQ were performed to all participants. My question is: is there any ASD individuals below the cut-off of these two scales? or any TD participants above the cut-off. the authors should consider excluding the ASD individuals below the cut-offs and TD individuals above the cut-offs from the data analysis.

This has been addressed in Question 8 of the “Essential Revisions.”

10. Please provide more details about how the TD participants were recruited?

We have now included this information (P13, “These subjects were recruited by flyers posted throughout Houston”). Importantly, we must note that we did screen for and exclude siblings of individuals with ASD.

11. IQ was available for a subset of the ASD participants: How many of them have IQ scores? Is there any particular reason that the other ASD participants did not have IQ scores? How the IQ was measured? using Wechesler or Raven's test? Was the IQ measured for the TD group?

This has been addressed in Question 8 of the “Essential Revisions.”

12. The authors could provide direct comparisons of thresholds and visual weights between two groups in the result section of Experiment 1.

We have expanded the Results section of Experiment 1 to more explicitly make the comparisons suggested by the reviewer (P4, “Overall, subjects with ASD […] with visual thresholds being equal in control and ASD across all reliability levels” and P4, “Measured visual weights were also not different between groups at any reliability (F(2, 114) = 1.11, p = 0.33)”).

13. Errors bars in Figure 1E and 1H were not very obvious. The authors could consider using simpler markers, such as "+" (i.e., short lines) for simultaneously displaying horizontal and vertical error bars.

We thank the reviewer for this suggestion. These figures have been modified, displaying simultaneously horizontal and vertical error bars. For these to be visible, instead of plotting standard error of the mean (SEM), we now plot 95% Confidence Intervals (CIs). We have also rendered the individual subject data (scatter plot) transparent, as to emphasize the error bars. The figure caption has been modified to reflect the change in illustration.

14. It should be "As for the case of auditory disparities, …" instead of " As for the case of spatial disparities, …" for the first sentence of the second paragraph after Figure 3.

This paragraph describes the explicit common cause reports during temporal disparities. The sentence highlighted by the reviewer is attempting to convey that we performed the same analyses as for spatial disparities (above). To reduce the chance of misunderstandings we now start this sentence “Analogous to the case of spatial disparities…”(P9).

Reviewer #2:The paper consists of 4 interesting experiments examining multisensory processing in autism spectrum disorder. The first experiment shows that participants with ASD perform similar to controls in cross-model integration, a conceptual replication of earlier findings from this group. However, the subsequent experiments reveal some intriguing differences between the groups in terms of how they use explicit and implicit information in evaluating if auditory and visual information comes from a common source or distinct sources. The authors propose a model that aims to explain the seeming dissociation between explicit and implicit reports of the two groups. The strength of this work is that the experiments are very interesting and report interesting results regarding audio-visual integration for spatial discriminations in both typical individuals and people with ASD. The comparison between explicit and implicit reports is very interesting. In terms of weaknesses, the dissociation between explicit and implicit is not convincing, and the stress on group differences puts an emphasis on, at best, marginal effects, which the modelling does not explain. For example, an alternative account that is consistent with all the data presented is that there are individuals with ASD who are somewhat poorer auditory discriminators, resulting in the bias effects and broader disparities. These individuals would be less likely to commit to an explicit "single source" statement in line with their reduced auditory localization skills.The dissociation between explicit and implicit is not convincing, and the stress on group differences puts an emphasis on, at best, marginal effects, which the modelling does not explain (the strongest linearity on ASD's curve in Figure 2 – is not captured in the modelling in Figure 4) For example, an alternative account that is consistent with all the data presented is that there are individuals with ASD who are somewhat poorer auditory discriminators and they impacted overall performance in Experiment 2, resulting in a larger bias effect, and also somewhat broader in disparities. These individuals would be less likely to commit to an explicit "single source" statement, which is quite committing, in line with their reduced auditory localization skills. The authors should at least address this alternative account, and present auditory discrimination curves of Experiment 2's participants.

We thank the reviewer for this very interesting set of comments and for proposing an alternative account.

In Figure S1 (novel addition during the revision) we plot the visual and auditory psychometric functions for all participants in Experiment 2 that also participated in Experiment 1 (panel A). 80% of participants in Experiment 2 also participated in Experiment 1, and we have confirmed that eliminating the 20% of subjects who did not participate in Experiment 1 does not change the conceptual results from Experiment 2. More importantly, and directly addressing the reviewer’s alternative explanation, as shown in panels B, C, and D, the thresholds, biases, and r-squared values were no different between the ASD and control cohorts. In turn, we can rule out the reviewer's alternative account. In addition to adding Figure 2 – supplement figure 1 in supplementary materials, we have modified the main text to reflect this analysis (P6, “To confirm that the larger […] across visual and auditory modalities, and for all reliabilities”).

The reviewer also suggests that the difference in causal inference between ASD and control subjects may be marginal. To ascertain whether the effect reported (i.e., individuals with ASD showing anomalous causal inference) is a robust one, we performed a conceptual replication and extension, as detailed in reply to Question #2 of the “Essential Revisions.” The results were replicated, suggesting that individuals with ASD outweigh integration relative to segregation when performing causal inference independent of sensory domain (see Figure 2 – supplement figure 2).

On the modeling front, we acknowledge that for the aggregate subject, the model is not able to completely capture all aspects of the data. However, our goal with the aggregate subject fitting was to understand what parameters could explain the overall difference between ASD and controls without losing interpretability (for example, we know from Experiment 1 that their sensory uncertainty is not different at a population level). Thus, we constrained the model such that only the choice and the causal inference parameters were allowed to vary. Further, it is not surprising that the aggregate data deviates somewhat from a causal inference model, since we combined the data from multiple subjects (with their own idiosyncrasies) in a largely model-agnostic way so as not to bias our results toward the causal inference model. On the other hand, the single subject fits are good, as we highlight in the reply to Question #2 of the “Essential Revisions” and in Figure 4 – supplement figures 3-6. Further, and most importantly, we have now also tested models where the choice and sensory uncertainty parameters are free to vary, while keeping p_common_ fixed (Alternatives A and B in Question #2, above, as well as Author response image 1). These models perform worse in accounting for implicit and explicit causal inference in ASD than the model where sensory uncertainty is fixed and p_common_ is allowed to vary, as quantified by AIC (see Figure 4 – supplement figure 7 – where C is fixed to 1 or 2, and Figure 4 – supplement figure 2, where C is fixed to the value obtained in control subjects).

The model does not account for the data point of individuals with autism being pulled by a reliable visual blob 24 degrees away, which was the main point in Figure 3.

We acknowledge that the aggregate subject model does not account for the data points at extreme disparities, and attribute this to two reasons.

First, the aggregate model for the ASD subjects is very constrained such that only the choice and causal inference parameters are allowed to vary relative to the control. Further, it is fit to all three visual reliabilities simultaneously. We used such a constraint since we wanted to drive the intuition about what parameters may drive the effect between ASD and control subjects. We show (see above, reply to Question #2 in “Essential Revisions”) that by removing this constraint we can capture the aggregate data better (Figure 4 – supplement figure 12, 13), but this leads to a loss of interpretability (parameters trading-off, and differences in sensory uncertainty not being supported by the empirical observations).

Second, the inability to capture the leftmost point arises from an asymmetry in the bias between the right-most and left-most disparities. This is due to combining data across subjects and is not explainable by a causal inference model (whose purview is the individual subject). In the original submission we summarized the quality of the individual subject fits using explainable variance explained (EVE). In Figure 4E you can see that the individual fits are in fact very good, explaining 80% of the EVE. In this revision we have now also included illustrations of the single subject fits (Figure 4 – supplement 3-6). Again, we must highlight that these fits are across all tasks and reliabilities, and not fine-tuned to account for responses in a single experiment.

Overall, we point out that our model for the ASD subjects deviates from the data in the direction of controls, i.e., is conservative. The raw data suggests that we are more likely to be underestimating than overestimating p_common_ for ASD subjects, and hence the difference to controls, further strengthening our overall conclusion.

Overall the authors ignore more prominent aspects of the data (e.g. higher overall bias in autism in Figure 2) for points they want to make (non linearity larger in autism than in controls).

We thank the reviewer for this comment and have modified the text to more clearly emphasize the overall differences in bias (P6, “Overall, individuals with ASD showed a larger bias (i.e., absolute value of the mean of the cumulative Gaussian fit) in auditory localization than the control group (see Figure 2A and Figure 2B, respectively, for control and ASD cohorts; F(1, 34) = 5.44, p = 0.002)”.*)*

However, we must highlight that the differences in bias exist at particular cue disparities (see Figure 2 and Figure 2 – supplement figure 1 and 2) and our goal here was to attribute well known differences in multisensory behavior (i.e., biases) to an underlying computation. Systematic biases occur when observers operate under an incorrect internal model. Thus, there being differences in biases is expected (at particular cue disparities) under causal inference. Further, notice that there is no difference in bias when no internal model is required, as in Experiment 1 (Figure 1 and Figure 2 – supplement figure 1 and Figure 3 – supplement figure 1). We do not ignore aspects of the data to make points we want to make, but instead focus on elements of the data that inform our understanding of the underlying computation (the biases being due to using incorrect internal models).

Reliability – is a confusing term. The stimuli are reliably presented, but the information the perceivers derive regarding their position is less reliable when stimuli are small.

We thank the reviewer for highlighting that “reliability” can be a confusing term. We now use this term to refer to the reliability of the information in the stimulus, or the reliability of the visual or auditory cue to avoid potential misunderstandings.

Figure 1f, g – I had difficulties understanding. I assume that the dashed lines should be to the right of the solid lines, which is the case for "high-reliability" blob, but why is it switched for the low reliability case? In both sample participants (f and g) and I wonder why the bias is larger (larger distance between dashed and matched solid plot, in both participants) for low versus intermediate size (reliability) blobs. If this is the actual result – it needs explanation.

We apologize, this was not clear in the text. Auditory thresholds were equal to visual thresholds at the intermediary reliability (shown in Figure 1D). At the high-reliability setting, visual thresholds were smaller than auditory ones. And in the low-reliability condition, visual thresholds were higher than auditory (Figure 1D). Thus, if participants are integrating cues in line with optimal cue combination, in the case of visual stimuli being highly reliable, participants' reports should be ‘pulled’ by visual location. Instead, in the low reliability condition, participants’ reports should be ‘pulled’ by the auditory location, given that it’s the most reliable one. At the intermediary reliability, both cues should influence the final report about equally. The example participants depicted in Figure 1F and 1G behave as predicted. The x-axis in the original version of the manuscript “stimulus location” was a misnomer, and likely the origin of the confusion. We apologize, this should have been “mean stimulus location” (given that it takes into account the relative location of the auditory and visual stimulus). Thus, as expected, when the reliabilities of the stimuli match (i.e., intermediary visual reliability), the dashed and solid line should be at the same location, and their slope should be maximal close to when the mean stimulus location (x-axis) is equal to zero. Instead, when visual reliability is high, the dashed curve should be to the right of the solid curve (indicating visual capture). When visual reliability is low, the dashed curve should be to the left of the solid curve (indicating auditory capture).

We have corrected the label of the x-axis in Figure 1F and 1G, and modified the text for clarity.

Figure 2 – the main observation is that the bias in autism is larger. Perhaps this group difference stems from this group being somewhat poorer auditory spatial discriminators than their 15 age-matched controls in the experiment. If their auditory discrimination is poorer we would expect an overall larger bias, and perhaps also across a broader range of audio-visual disparities.

There was no difference in visual or auditory discrimination performance among the subjects in Experiment 2, see Figure 2 – supplementary figure 1 and reply to comments above.

Importantly, this is probable account, since this is a smaller population than in Experiment 1 – and their discrimination thresholds are not addressed. Importantly – I could not figure out the overlap in participation across the various experiments. In experiment 1 matched performance was only obtained when 6 participants with ASD were excluded. In Experiment 3 (24 participants originally) – they also excluded a large subgroup, whose behavior was different. Here the group is initially small so variability across participants was not discussed.

We apologize for not providing this information in the initial manuscript. We have now added Supplementary File 1 (controls) and Supplementary File 2 (individuals with ASD) to detail which participants took part in the different experiments (see response to Reviewer #1 and “Essential Revisions”). As mentioned above, there was no difference in unisensory performance among participants taking part in Experiment 2.

The strongest point for the claim of too broad integration is the bottom left point – where high reliability blob has an effect that even increases when the visual blob is presented 24 degrees apart. This point is hard to reconcile (and is not reconciled by the model proposed in Figure 4 either). The authors should show that it is a reliable data point – perhaps by showing single subject data.

First, we would like to point out that while we agree that this one data point is particularly compelling in a qualitative way, our conclusions would hold even in its absence. Also, as we have addressed above, individual fits are good (see Figure 4E) and we have now included example single subject data; 2 per experimental cohort (Figure 4 – supplementary figures 3-6). The aggregate data fits are (1) very constrained (fitting to multiple tasks and reliabilities while solely varying choice and inference parameters) and (2) could be improved at the expense of losing interpretability (see above). Most importantly (3) the quality of the aggregate data fits speaks to inter-subject heterogeneity, and not the ability of the causal inference model to account for individual responses (which is quantified in Figure 4E and illustrated in Figure 4 – supplementary figures 3-6). The point to the lower left, 24 degrees in Figure 2B, is reliable, as shown by the S.E.M.

In addition to the individual subject fits, we have now also included a completely new experiment (heading discrimination during object-motion) in the supplementary materials (Figure 2 – supplementary figure 2), demonstrating again an impairment of causal inference in ASD (see replies to “Essential Revisions” above).

In experiments 3 and 4 the specific instructions are crucial – are participants asked to press a specific button if they are perceived as coming from the same source? Or press a button if they are perceived as coming from 2 separate sources. Here phrasing may have affected the decisions of individuals with autism. In order to dissociate between these 2 options it would have been nice to have a third option "don't know". If participants with autism tend to say to be less decisive they would tend to commit to a single source. This account may be explained by being somewhat implicitly poorer localizers.

We thank the reviewer for this question and agree that in the future it may be interesting to allow for a third – non-committal – option. We have added to the text the explicit instructions that were given (see reply to Question #5 in the “Essential Revisions”). We do not believe that the specific phrasing drove the explicit effects we report, given that the phrasing was different for the spatial (Experiment 3) and temporal task (Experiment 4), while their reduced tendency to report common cause was shared across experiments. Further, if the phrasing does play a strong role (something we cannot be sure of in this experiment), and a stronger role in ASD relative to controls, this would simply provide an alternative explanation for the categorical bias that we found in the explicit task (i.e., a compensatory mechanism), while leaving our conclusions about the differences in p_common_ unchanged. It would therefore add and not detract or oppose the current findings.

If you have discrimination functions of the specific subgroups that took part in Experiments 2-3 (since they all participated in Experiment 1 – right?) – please show them or report discrimination skills for these subgroups, since this is the relevant control-ASD matching.

We thank the reviewer for highlighting that this important control was missing. The discrimination functions for the subgroup participating in Experiment 2 are including as Figure 2 – supplementary figure 1, while these functions for the subgroup participating in Experiment 3 are included as Figure 3 – supplementary figure 1. The cohort of control and ASD participants taking part in Experiments 2 and 3 were no different from each other with regard to visual or auditory localization performance (as is also true of Experiment 1). This important information has been added to the text (P6, “To confirm that the larger biases […] and for all reliabilities”. And, P7, “See Figure S3 for the unisensory discrimination performance […] in this explicit causal inference judgment experiment”).

Re modelling and Figure 4 – It is difficult to follow the model – perhaps label the model parameters in the diagram of Figure 4a.

We thank the reviewer for their suggestion. We have updated the model figure to increase clarity. We have separated the generative models for the implicit and explicit task, and have included the parameter associated with each step of the generative model.

Reviewer #3:In this paper Noel et al., use a combination of psychophysical experiment and computational modeling to examine the differences in behaviour between participant on the Autism Spectrum Disorder and control participants when dealing with multi-sensory stimuli (e.g. audio-visual). It is well known that ASD subjects tend to differ in how they combine such stimuli, and it has previously been suggested that this may be due to a difference in the tendency to perform causal inference.The study indeed finds that while ASD participants had similar ability to combine cues when unambiguously from the same source, they differed in the tendency to combine them when unclear if necessary to combine. In contrast when asked to explicitly indicate whether stimuli originated from the same source (and therefore should be combined) they tended to under report.While the experiments are in themselves very standard, the paper relies on computational modeling to differentiate the possible behavioural effects, using advanced Bayesian statistical methods.These results confirm existing ideas, and build on our understanding of ASD, while still leaving many questions unanswered. The results should be of interest to anyone studying ASD as well as any other developmental disorders, and perception in general.I enjoyed reading this paper, although the model fitting procedure especially was not clear to me. How were the sensory parameters fitted? By my count more than 20 parameters were fitted (Supp. File 1) for the aggregate subject through the slice sampling, is that correct? Was this also done for individual subjects? I would be nervous about fitting that many parameters for individual subject data. What was done to ensure convergence?

We thank the reviewer for his question. First of all, we want to emphasize that we perform full Bayesian inference over all model parameters given the empirical data. What this means is that we are computing a joint distribution over all parameters that captures all of our knowledge about these parameters given the subject responses (we do this by slice sampling but that is just a technical detail). Importantly, if some parameters are not constrained by our data, then the posteriors over them will be very wide, in the extreme case simply corresponding to the prior distributions that reflect our knowledge about them in the absence of any new data. Or if there is a degeneracy such that e.g., only the sum of two parameters is constrained, but not each of them individually, then this will also manifest itself in very wide individual error bars. Importantly, when reporting our estimates and confidence intervals about the parameters that we do care about, e.g. p_common_, we account for the uncertainty in all the other parameters.

Regarding the number of parameters, yes, each subject had 20 parameters characterizing their responses. However, this was across four tasks (unisensory and small disparities discrimination, multisensory discrimination, implicit causal inference, and explicit causal inference) and three visual reliability levels. Further, the prior over the parameters reduces the effective degrees of freedom, such that several parameters have their values tied together (unless the observed experimental data provides evidence to the contrary). We have chosen this approach to allow for variance in parameters (e.g., lapse parameters) across experiments that were conducted on different days, but a-priori assuming they are the same. Therefore, for the aggregate subject, when we fit the control data, we are effectively fitting 12 parameters for the implicit task and 11 parameters for the explicit task (subset relevant for each task). Also, since the model was fit to multiple reliabilities, the prior parameters that were shared across reliabilities were constrained by data from all three reliabilities. For the ASD aggregate subject, either only the choice parameters were varied, or the choice parameters and p_common_ were varied (relative to the aggregate control subject). We followed a similar procedure for the fit to individual subjects, but additionally used subject specific data from the different experiments to constrain parameters (notably Experiment 1 and estimates of sensory uncertainty).

We have now summarized the model fitting procedure as a flowchart in the supplementary (Figure 4 – supplementary figure 1).

Was any model comparison done? Might be better to include a list or figure showing the different steps of the model fitting.

In this revision we have considered a number of different models that could explain the difference between the aggregate control and ASD subject. These alternative models are:

A. Forced fusion (all parameters are free, except for C, which is fixed to 1).

B. Forced segregation (all parameters are free, except for C, which is fixed to 2).

C. Lapse rate, p_common_, p_choice_ are free with uniform lapse bias.

D. D1) Implicit task: Lapse rate and bias and p_choice_ are free

D2) Explicit task: Since p_choice_ trades off against p_common_ for the explicit task only lapse rate and bias are free

We quantify the goodness of fit by AIC and contrast these alternative models to that presented in the main text (Figure 4 – supplementary figure 7). These models all perform worse than the one where p_common_, but not sensory uncertainties are allowed to vary across the control and ASD cohorts. See reply to Question #1 of the “Essential Revisions” and the main text (P9, 10) for further detail.

I also worry that the model is over specified with both a lapse rate and a lapse bias. From my understanding the lapse rate specifies when subjects (through lack of concentration or otherwise) fail to take trial stimuli into account and therefore go with their prior. In other studies this prior may be identical to the prior over spatial range, or may be a uniform discrete distribution over the bottoms available for response.Maybe the variables are constrained in ways that I did not understand, but with just a binary response (Left/Right) the model can largely incorporate any bias to a large set of possible parameter values of lapse rate and bias. I.e. that the model is over specified. That would also explain the wide range of values for the fitted parameters in Figure 3.I think this should really be investigated before the results can be trusted.Looking at Figure 4E and F makes me hesitant about trusting the results.Authors also acknowledge that the lapse bias and P combined are too closely entwined to really be well separated in the explicit temporal experiment. Maybe for that reason it would also be useful to test a simpler model without lapse bias?

Our prior over the lapse bias (i.e., the bias in the response given a lapse) peaks at 0.5. Thus, the model implicitly assumes no lapse bias, unless supported by the data. To further confirm that the empirical results do support the presence of a lapse bias (and thus require it in modeling), we have now fit a model with a uniform lapse bias for the aggregate subject (Alternative C in Response #1 to “Essential Revisions”). This model had a worse AIC than that allowing for a lapse bias (see Figure 4 – supplementary figure 7), suggesting the data supports the presence of a lapse bias. Regardless, the observation that individuals with ASD have a larger p_common_ than controls holds for both models with and without a lapse bias.

Similarly, we obtained better model fits to individual subject data while allowing for the possibility of lapse biases. Only 6 subjects had a smaller AIC for the model without (vs. with) lapse bias. For those subjects, the estimated p_common_ was not statistically different from the model with the lapse bias (p=0.68). Therefore, while using a model without a lapse bias does not change our conceptual results regarding p_common_, we believe that incorporating a lapse bias and then marginalizing out any potential contribution from it allows us to better estimate the actual contribution of the variables of interest.

We understand the reviewer’s concern that the model may be over-specified, particularly when fitting to a limited dataset (i.e., individual subjects). However, as described above, this is not a problem since we perform full Bayesian inference over all model parameters while fitting data from 4 tasks and 3 different reliabilities, simultaneously. Further, any under-constraint in the model would manifest as correlated posteriors, and large uncertainties in the parameter estimates. While the heterogeneity across subjects in Figure 4F and G is high, the confidence intervals over the subject specific parameters (the error bars around individual dots) is not, indicating that this subject-to-subject variability reflects actual differences between people.

I find it mildly confusing that D refers to a Left/Right response in the implicit task, and Common/Separate in the explicit task. Maybe better to use separate symbols? D is fine for 'decision' but in places in the text it is instead referred to as 'trial category' which is vague. I also don't really think D is needed in the generative model in Figure 4 as it is not really causing the subsequent variables C or Sa.

We want to clarify that D is not the response but the category of the trial. We model the subject as using a common perceptual framework across tasks: observers generate beliefs about the locations (s) that generated their observations (o). This belief is related to a belief over the trial category (D) that generated the observations in the trial. The response, which we refer to as R, is generally whichever belief is greatest (excluding lapses). This response minimizes the expected loss according to Bayesian decision theory. The reviewer is right that the response does not fit into the generative model, but the trial category does, since the experimenter has to infer the trial category in every trial.

Does eLife not require the reporting of effect sizes (e.g. eta^2^ or Cohen's d)? It would be good to include these.

We thank the reviewer for this suggestion. Effect sizes have now been added throughout the manuscript.

The plots in Figure 3 mostly look like shifts up for ASD relative to controls. The authors might want to fit a model with a positive bias, i.e.a*N(mu,sd^2^)+bmay fit better (could do model comparison) and just show difference in b. This is just a suggestion though, but it may be cleaner for their argument.

We thank the reviewer for his suggestion and have attempted this modelling approach. We have added a new supplementary figure (Figure 3 – supplementary figure 3), showing that this model performs worse than the causal inference in terms of AIC. We have amended the text to reference this attempt (P22, Lastly, as a contrast to […] than the functional form).

In the Discussion, while divisive normalisation is one way to achieve the marginalisation needed for Bayesian causal inference, there are other ways to achieve it (Cuppino et al., 2017, Yamashita 2013, Yu et al., 2016, Zhang et al., 2019). It would be good to acknowledge this.

The reviewer is entirely correct and pointing toward prior work implementing neural networks of causal inference is extremely relevant. We have reviewed the reports cited by the reviewer; two of them make explicit reference to normalization in their modeling efforts, while the others do not (Zhang et al., 2019, for example, relying on the finding of “congruent” and “incongruent” cells). We have amended the discussion to acknowledge that divisive normalization is only one of many possible ways of achieve marginalization (P12, “The juxtaposition between […] Yamashita et al., 2013; Yu et al., 2016).

Equation 5 and 6, 38 are misleading. Likelihood is a function of Sa/Sv, so would be better to write as l(Sa)=N(Xa;Sa,Sv)

We thank the reviewer for their suggestion. We have added the likelihood function definition as suggested by the reviewer.

Equation 9: is D either 1 or 2? Or 1 or -1?

D is 1 or -1 in the implicit task where it refers to the side on which the tone is inferred, and D is 1 or 2 in the explicit task, where it refers to the number of causes inferred for the observations. For clarity, we have now separated the two using D_imp_ and D_exp_ where D_imp_ is -1 or 1 and D_exp_ is 1 or 2.

Detail: maybe use different symbols for lapse rate and lapse bias? I find λ and lambdar confusing. How about P_lapse_ for the lapse rate to emphasise that it is a probability? P_common_ is already a fitted variable that is also a probability of a Bernoulli distribution

As suggested by the reviewer we have replaced λ and λ_r with p_lapse rate_ and p_lapse bias._

Page 5 (pages of the pdf):" …ASD did not show impairments in integrating perceptually congruent auditory and visual stimuli."– " …ASD did not show impairments in integrating perceptually congruent (and near-congruent) auditory and visual stimuli."In experiment 2 there was a six degree discrepancy, so near-congruent seems appropriate.

The text has been amended as suggested by the reviewer.

Typos:"We perform the integral in Equation S5 for the implicit task by": should this be Equation 35?

Indeed, in the original version of the manuscript this should have been Equation 35. In the current version of the manuscript we have moved this section to the supplementary materials.

[Editors’ note: what follows is the authors’ response to the second round of review.]

Reviewer #1:1. Figure legends/captions of Figures 3 and 4 in the main texts lack detailed descriptions of the elements in the figures. For example, for Figures 3 and 4, what do those error bars represent? Standard errors or confidence intervals? In Figure 4B, are solid lines the model predictions and hollow points the observations? I believe this essential information would help readers better understand the figures.

We thank the reviewer for highlighting that we had missed this important information. In Figure 3, indeed, error bars represent standard error of the mean (SEM). In Figure 4B and C, error bars are 68% confidence intervals. Similarly, in Figure 4F and G, the individual subject-level error bars are 68% confidence intervals (equivalent to standard deviations under Gaussianity). In Supplementary File 4, we elaborate on how we obtain confidence intervals for individual (or amalgamated) subjects:

“We also obtained full posteriors over model parameters for the individual subjects by jointly fitting the model to all experiments with weakly informative priors. We modeled each observer population with a hierarchical model where an individual observer’s parameters are independent draws from the population parameter, parameterized as a Gaussian, i.e. θsubject ∼N(θpopulation,σpopulation2) where θ are the parameters of the causal inference model. We also approximated the posterior over the subject parameters as Gaussians which allowed us to analytically combine the individual posteriors to a combined Gaussian population posterior for further hypothesis testing and obtaining confidence intervals (CI). For the individual subject estimates, we plot 68%CI. We obtained the p-value for the difference between ASD and control subjects parameters using two methods: (a) Normal approximated analytical p-value which assumes that the sample variance is equal to the population variance, (b) Welch t-test which relaxes the above assumption. Both methods gave comparable p-values and we conservatively considered the higher p-value for significance testing.”

In Figure 4F and G group-level error bars are +/- 95% confidence intervals and hence non-overlapping bars indicate a statistical significance at p < 0.05. As the reviewer inferred, the “dots” in Figure 4B and C are data, and the lines are fits to this data. We have modified the figure captions (Figure 3 and 4) to include all this important information.

2. The data points in Figure 2A-B and Figure 3A-C are slightly different from those in Figure 4B-C. For example, in Figure 2B, the audio bias of 24 deg disparity is weaker than that of 12 deg disparity for the high visual reliability condition (dark brown lines and points); however, in Figure 4B left panel, the audio bias of 24 deg disparity is even larger than that of 12 deg disparity. I assume that the data points depicted in Figure 4B-C are the aggregate data for modeling, in which the data of some participants were not included? I notice that the authors have included which participants were included in the single-subject modeling, but was the aggregate data the same as what was used for plotting Figures 2 and 3? I find it a bit confusing at first sight, perhaps the author could check it again and/or mention the related information in the caption or the main text?

We thank the reviewer for highlighting that the difference in data between Figures 2/3 and Figure 4 can be confusing. The difference is due to the fact that while in Figures 2/3 data is averaged within subjects, then psychometric fits are performed at the single subject level (e.g., Figure 1B, C, F, G), and finally psychometric estimates are averaged across subjects, in Figure 4 data is amalgamated and averaged across all subjects, and a single psychometric fit is performed. We have added this information in the figure caption to Figure 4.

3. From lines 451-453 of merged files (Instead, differences between […] relative to control observers.), did the author imply that the model where p_common_ was freely estimated from the data was better, compared with the model where p_common_ was fixed (I guess it's the model in Figure 4 – supplement 2)? In other words, did the authors have two different models and conduct a model comparison here? If so, I think it's better to provide model comparison results. The question also applies to the texts from lines 460-461. Also, what is DAIC? Is it the difference of AIC between the full model (that allows p_common_) and the restricted model (that fixes p_common_ to a constant)? The authors should describe it somewhere in the main text.

Indeed, DAIC stood for the difference in AIC between two models. We have changed this nomenclature to ∆AIC, given the use of “∆” for difference throughout the text and in Figures 4 – supplement 2, 7, 12, and 13.

The models we compared in the main text were:

– For the implicit task, a model where only choice parameters (choice bias + lapse rate + lapse bias) were free to vary vs. a model where both choice and inference (p_common_) were free to vary.

– For the explicit task, a model where only lapses (rate and bias) were free to vary (given the impossibility to distinguish the choice bias from p_common_) vs. a model where both lapses and “p_combined_” were free to vary.

This is indeed close to what we present in Figure 4 – supplement 2, with the exception that we additionally add the sensory uncertainty in the supplement (which is not added in the main text given the empirical results demonstrating no difference between groups).

We have modified the text in the following manner to make this clearer:

“In the implicit task (Figure 4B, top panel), allowing only for a difference in the choice parameters (lapse rate, bias, and p_choice_; magenta) between the control and ASD cohorts, could only partially account for observed differences between these groups (explainable variance explained, E.V.E = 0.91, see Supplementary File 4). Instead, differences between the control and ASD data could be better explained if the prior probability of combining cues, p_common_, was also significantly higher for ASD relative to control observers (Figure 4D, p = 4.5x10^-7^, E.V.E = 0.97, ∆AIC between model varying only choice parameters vs. choice and inference parameters = 1x10^3^). This suggests the necessity to include p_common_ as a factor globally differentiating between the neurotypical and ASD cohort.”

And:

“For the explicit task, different lapse rates and biases between ASD and controls could also not explain their differing reports (as for the implicit task; EVE = 0.17). Differently from the implicit task, however, we cannot dissociate the prior probability of combination (i.e., p_common_) and choice biases, given that the report is on common cause (Figure 4A, see Methods and Supplementary File 4 for additional detail). Thus, we call the joint choice and inference parameter p_combined_ (this one being a joint p_common_ and p_choice_). Allowing for a lower p_combined_ in ASD could better explain the observed differences between ASD and control explicit reports (Figure 4C; EVE = 0.69, ∆AIC relative to a model solely varying lapse rate and bias = 1.3x10^3^). This is illustrated for the ASD aggregate subject relative to the aggregate control subject in Figure 4D (p = 1.8x10^-4^)”

4. The authors should be more specific about the tests they used to compare model parameters between groups and those correlational analyses. What type of tests did the authors use, parametric (i.e., Welch t-test, Pearson correlation) or non-parametric (i.e., Mann-Whitney, Spearman correlation, or permutation methods)? Particularly for the comparison of p_combined_ (Figure 4G), would the result be different when a non-parametric test was used if the test used in the current revision was parametric? I suggest the authors take more robust approaches given that the distributions of the model parameters seemed not quite Gaussian.

Regarding Figure 4G, as we indicate in Supplementary File 4, we conducted both parametric and non- parametric t-tests. We then conservatively considered the higher p-value. The relevant piece of text is:

We obtained the p-value for the difference between ASD and control subjects parameters using two methods: (a) Normal approximated analytical p-value which assumes that the sample variance is equal to the population variance, (b) Welch t-test which relaxes the above assumption. Both methods gave comparable p-values and we conservatively considered the higher p-value for significance testing.

For the correlations, we performed Type II regression (indicated in the text). This approach appropriately considers that both measures being correlated are noisy estimates, and thus each of the two variables regressed are first transformed to have a mean of zero and a standard deviation of one (Ricker, 1973).

5. What is α and ν in Equation 5 and 6, please define them in the text. Also, it would be better if the authors give a short introduction to the meaning of lapse rate, lapse bias, etc., when mentioning them for the first time. Given that many readers are not very familiar with computational modeling, they may not intuitively understand what these parameters represent.

We have modified the text in order to introduce α and ν, as well as give a short introduction to the meaning of lapse rate, lapse bias, and prior. The text has been modified in the following manner:

We assume that subjects have a good estimate of their sensory uncertainties (over lifelong learning) and hence the subject’s estimated likelihoods become,

l(Sa)≡p(Xa|Sa)= N(Xa;Sa,σa2)(Equation 5)

l(Sv)≡p(Xv|Sv)= N(Xv;Sv,σv2)(Equation 6)

where Sa and Sv denote the inferred location of auditory and visual stimuli.

And:

First, sensory parameters: the visual and auditory sensory uncertainty (i.e., inverse of reliability), as well as visual and auditory priors (i.e., expectations) over the perceived auditory and visual locations (mean and variance of Gaussian priors). Second, choice parameters: choice bias (p_choice_), as well as lapse rate and bias. These latter two parameters are the frequency with which an observer may make a choice independent of the sensory evidence (lapse rate) and whether these stimuli-independent judgments are biased (lapse bias). Third, inference parameters: the prior probability of combination (p_common_; see Methods and Supplementary Files 3 and 4 for further detail).

6. The D in DAIC from line 462 is in another font.

We thank the reviewer for noticing this typo. As indicated above, we have changed this nomenclature to “∆AIC”.

7. I apologize in advance if it's my mistake but I failed to find Supplementary Text 1 mentioned in lines 430, 451, and 459. Where could I find it?

This is our mistake, we apologize. Supplementary Text 1 is now Supplementary File 4.

Reviewer #2:The authors have adequately addressed my comments.The strong aspects of the results are better clarified, and the overlap between participants across experiments is also clear. Further, the authors do not make claims that are not directly supported experimentally.The limitation of a somewhat small (<20) number of participants per group in important experiments is still a drawback, given participants' variability, particularly in the ASD group. Yet, I believe that the main results hold.

We thank the reviewer for helping us strengthen and clarify the results in this manuscript. We agree that within which experiment, the sample sizes were of moderate size. We have amended the discussion to acknowledge this limitation:

However, it must be acknowledged that while the overall number of participants across all experiments was relatively large (91 subjects in total), our sample sizes within each experiment were moderate (~20 subjects per group and experiment), perhaps explaining the lack of any correlation.

The strongest aspects of the study are the direct results, rather than the modelling:Experiment 1: audio-visual integration is intact in ASD 2. yet multisensory behavior is atypical (in the current experimental protocol) – ASD participants tend to favor source integration, as manifested by their cross-modal bias in localization even when visual and auditory signal are separable from a sensory perspective. Though both groups tend to over integrate, this is more salient and tend to span a broader distance in ASD. 3. Explicit reports have an opposite tendency – individuals with ASD were less likely to report a common cause for the two stimuli. Given the adequate direct measures of ASD cue integration with a small audio-visual distance (performance in Experiment 1) these results suggest a specific atypicality in cause attribution.I also find the difference between spatial and temporal integration very interesting. Temporal and spatial groups differences in explicit attribution of a common source merits some additional discussion.

We agree. We have expanded the discussion in the following manner:

“This has previously been observed within the temporal domain (Noel et al., 2018a, b), yet frequently multisensory simultaneity judgments are normalized to peak at ‘1’ (e.g., Woynaroski et al., 2013; Dunham et al., 2020), obfuscating this effect. To the best of our knowledge, the reduced tendency to explicitly report common cause across spatial disparities in ASD has not been previously reported. Further, it is interesting to note that while “temporal binding windows” were larger in ASD than control (see Feldman et al., 2018), “spatial binding windows” were smaller in ASD relative to control subjects. This pattern of results highlights that when studying explicit “binding windows”, it may not be sufficient to index temporal or spatial domains independently, but there could potentially be a trade-off.”